# Multimodal Large Language Models for Inverse Molecular Design with Retrosynthetic Planning

**Gang Liu**[1]*, **Michael Sun**[2]*, **Wojciech Matusik**[2], **Meng Jiang**[1], **Jie Chen**[3]
[1]University of Notre Dame    [2]MIT CSAIL    [3] MIT-IBM Watson AI Lab, IBM Research
{gliu7, mjiang2}@nd.edu,  {msun415, wojciech}@csail.mit.edu,
chenjie@us.ibm.com

## Abstract

While large language models (LLMs) have integrated images, adapting them to graphs remains challenging, limiting their applications in materials and drug design. This difficulty stems from the need for coherent autoregressive generation across texts and graphs. To address this, we introduce Llamole, the first multimodal LLM capable of interleaved text and graph generation, enabling molecular inverse design with retrosynthetic planning. Llamole integrates a base LLM with the Graph Diffusion Transformer and Graph Neural Networks for multi-conditional molecular generation and reaction inference within texts, while the LLM, with enhanced molecular understanding, flexibly controls activation among the different graph modules. Additionally, Llamole integrates A* search with LLM-based cost functions for efficient retrosynthetic planning. We create benchmarking datasets and conduct extensive experiments to evaluate Llamole against in-context learning and supervised fine-tuning. Llamole significantly outperforms 14 adapted LLMs across 12 metrics for controllable molecular design and retrosynthetic planning. Code and model at `https://github.com/liugangcode/Llamole`.

## 1 Introduction

The potential of LLMs for molecular discovery has been actively explored (Jablonka et al., 2023). However, LLMs struggle in the chemical domain, exhibiting poor generation quality and planning capability (Guo et al., 2023). This is due to the unique graph structures of molecular data, which are challenging for LLMs that typically handle sequential texts.

Inverse molecular design requires LLMs to be controllable for generating molecular structures that meet multi-property and synthesizability requirements (Chen et al., 2020; Gao et al., 2021). These requirements can be detailed as questions for LLM input, as shown in Figure 2. Answering these questions demands a comprehensive understanding of molecular structures and their relationship to properties. However, sequence-based LLMs struggle with this because they are pre-trained or fine-tuned solely on text representations of molecules, e.g., SMILES (Weininger, 1988). To illustrate this, we investigate 14 LLMs for molecular generation in Figure 1 across 10K drug and material questions: ten using in-context learning (ICL) and four with supervised fine-tuning (SFT). LLMs generate molecular structures based on the questions and their properties are obtained through oracles Details of the experimental set-ups and results can be found in Section 5. In summary, even the best LLMs perform worse than GraphGA (Gao et al., 2022), a simple yet effective graph-based method, in designing molecules with satisfactory properties.

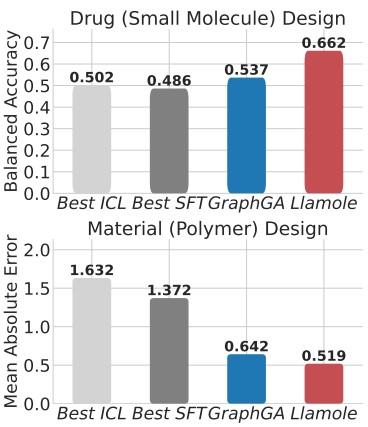

Figure 1: Comparison of Controllability: Results are averaged from the best numbers from Table 1.

---

*This work was done while GL and MS interned at the MIT-IBM Watson AI Lab, IBM Research.

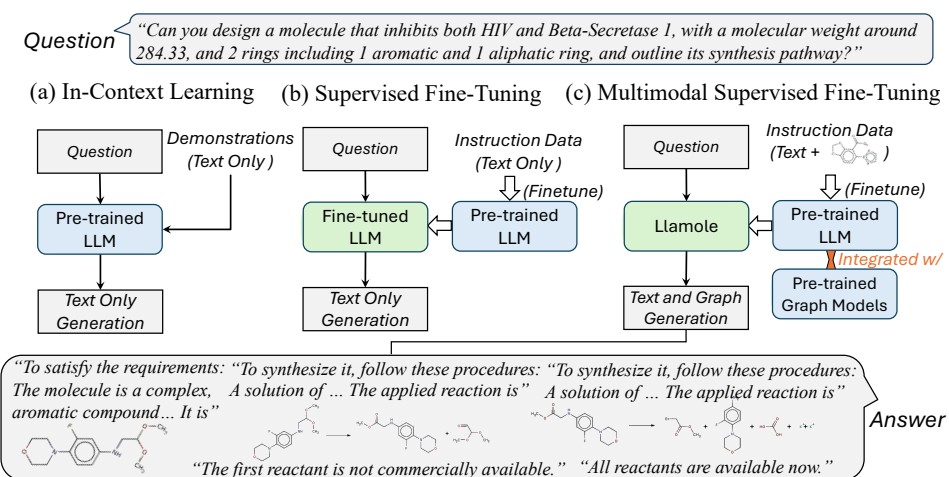

Figure 2: Three LLM-based methods for molecular design. The question outlines requirements for properties, structures, and synthesis, addressed as follows: (a) In-Context Learning and (b) Supervised Fine-Tuning use text-only data for demonstrations and instruction tuning, respectively. (c) The proposed Llamole uses graph-text multimodal data to fine-tune the LLM, integrating parameter-frozen graph models for interleaved text and graph generation with reaction inference.

As illustrated in Figure 2, practical answers for molecular design are more complex than what can be achieved by using graph methods or LLMs alone. The generation begins with a paragraph describing the intended molecule for multi-conditional generation, followed by retrosynthetic planning, detailing each synthesis step—one reaction per paragraph—in reverse order, from the target molecule to purchasable reactants. Thus, multimodal LLMs (MLLMs) are essential, with LLMs handling text generation and graph models managing molecular design.

In this work, we propose the multimodal **L**arge **la**nguage model for **mole**cular discovery (**Llamole**). As shown in Figure 2 (c), the model seamlessly integrates LLMs and graph models within a multimodal autoregressive framework, enabling the interleaved generation of text, molecules, and reactions. It predicts the next token across both word and chemical spaces, framed as multi-class prediction tasks for word vocabulary, atom/bond types, and reaction templates. For retrosynthetic planning, Llamole integrates A* search to efficiently identify synthesis pathways for the designed molecule.

To implement Llamole, we augment a base LLM with two pre-trained graph modules: the Graph Diffusion Transformer (Graph DiT) for multi-conditional molecule generation (Liu et al., 2024c) and a GNN for reaction template prediction. The base LLM controls the generation flow using a trigger-query-prediction approach with two sets of trigger tokens for the Graph DiT and GNN, respectively. Upon predicting a trigger token, one or a few query tokens summarize the prior text as vectors, activating the corresponding graph modules and generating molecules or predicting reaction templates. Afterward, the base LLM can resume text generation, aided by a graph encoder that encodes the previously generated molecule. In retrosynthetic planning, the LLM computes heuristics to efficiently assist the A* search in navigating the vast reaction space for multi-step generation.

Our work has several highlights. First, Llamole is the first MLLM capable of inverse molecular design with the interleaved generation of text and graphs. Second, we curated a dataset along with fine-tuning instructions to benchmark complex yet realistic molecular design outcomes, including human conversation. Third, we present compelling experimental results that demonstrate the competitiveness of Llamole against 14 LLMs and GraphGA, as shown in Figure 1. With details in Tables 1 and 2, Llamole improves LLM performance by up to 80.9% across 12 metrics for controllable molecular generation and increases the success rate for retrosynthetic planning from 5.5% to 35%.

## 2 PRELIMINARIES

### 2.1 AUTOREGRESSIVE LANGUAGE MODELING

Given a sequence of word tokens $W = \{w_1, w_2, \ldots, w_L\}$ of length $L$ from the vocabulary $\mathcal{W}$, LLMs parameterized by $\theta_1$ decompose the joint distribution as $p_{\theta_1}(W) = \prod_{i=1}^{L} p_{\theta_1}(w_i | W_{<i})$, where $W_{<i}$

represents the tokens preceding the $i$-th position. These models are optimized by minimizing the negative log-likelihood between their predictions and the empirical data distribution, resulting in:

$$\mathcal{L}_{\text{LM}} = \sum_i - \log p_{\theta_1}(w_i | W_{<i}). \tag{1}$$

## 2.2 MOLECULAR DESIGN WITH GRAPH DIFFUSION MODELS

Molecular graphs can be modeled through diffusion in discrete spaces (Austin et al., 2021; Vignac et al., 2022; Liu et al., 2024c). Given a one-hot encoded data point $\mathbf{x} \in \mathbb{R}^F$ with $F$ categories (e.g., a node or an edge), discrete models perform diffusion using a transition matrix $\mathbf{Q}$, where $[\mathbf{Q}^t]_{ij} = q(\mathbf{x}_j^t \mid \mathbf{x}_i^{t-1})$ for $i,j \in [1,F]$. The forward diffusion with $\mathbf{Q}$ is: $q(\mathbf{x}^t \mid \mathbf{x}^{t-1}) = \text{Cat}(\mathbf{x}_t; \mathbf{p} = \mathbf{x}^{t-1}\mathbf{Q}^t)$, where $\text{Cat}(\mathbf{x}; \mathbf{p})$ denotes the categorical distribution over $\mathbf{x}$ with probabilities given by $\mathbf{p}$. Starting from the original data point $\mathbf{x} = \mathbf{x}^0$, we have $q(\mathbf{x}^t \mid \mathbf{x}^0) = \text{Cat}\left(\mathbf{x}^t; \mathbf{p} = \mathbf{x}^0\bar{\mathbf{Q}}^t\right)$, where $\bar{\mathbf{Q}}^t = \prod_{i \leq t} \mathbf{Q}^i$. The forward diffusion gradually corrupts data points. When the total timestep $T$ is large enough, $q(\mathbf{x}^T)$ converges to a stationary distribution. The reverse process samples from $q(\mathbf{x}^T)$ and gradually removes noise. The posterior distribution $q(\mathbf{x}^{t-1} \mid \mathbf{x}^t)$ is calculated as $q(\mathbf{x}^{t-1}|\mathbf{x}^t,\mathbf{x}^0) \propto \mathbf{x}^t(\mathbf{Q}^t)^\top \odot \mathbf{x}^0\bar{\mathbf{Q}}^{t-1}$. Using a denoising model parameterized by $\theta_2$, this posterior can be approximated by $p_{\theta_2}(\mathbf{x}^{t-1}|\mathbf{x}^t,\mathbf{x}^0)$. For inverse molecular design with multi-property constraints, the denoising model can be optimized by minimizing the negative log-likelihood for $\mathbf{x}^0$:

$$\mathcal{L}_{\text{DM}} = \mathbb{E}_{q(\mathbf{x}^0)}\mathbb{E}_{q(\mathbf{x}^t|\mathbf{x}^0)}\left[-\log p_{\theta_2}\left(\mathbf{x}^0 \mid c_1, c_2, \ldots, c_M, \mathbf{c}_{\text{text}}, \mathbf{x}^t\right)\right], \tag{2}$$

where $M$ molecular properties are denoted by $\{c_i\}_{i=1}^M$, and the text embedding is $\mathbf{c}_{\text{text}}$. These conditions can be handled by Graph DiT (Liu et al., 2024c) without introducing additional predictors for guidance (Ho & Salimans, 2022).

## 2.3 ONE-STEP REACTION PREDICTION WITH GRAPH NEURAL NETWORKS

Retrosynthesis needs to predict the reverse of a synthetic reaction, which decomposes chemical products into reactants. A GNN parameterized by $\theta_3$ takes the product $G_{\text{product}}$ to predict the label $r \in \mathcal{R}$ in the reaction space $\mathcal{R}$. This label is interpreted as the template and determines the reactants. With the text condition $\mathbf{c}_{\text{text}}$, we minimize the negative log-likelihood of the label distribution $q(r)$:

$$\mathcal{L}_{\text{predictor}} = \mathbb{E}_{q(r)}\left[-\log p_{\theta_3}(r \mid \mathbf{c}_{\text{text}}, G_{\text{product}})\right]. \tag{3}$$

## 2.4 RETROSYNTHETIC PLANNING WITH A* SEARCH

Given molecules from the structure space $\mathcal{G}$, a subset $\mathcal{G}_{\text{avail}}$ represents available molecular structures that can be purchased as building blocks for synthesis. For any target $G_{\text{target}}$, one-step prediction of the reversed reaction may not yield reactants within $\mathcal{G}_{\text{avail}}$. Thus, retrosynthesis typically requires multi-step planning to find pathways from building blocks to the target in reverse order. The search space of chemical reactions can be navigated using A* on an AND-OR tree $\mathcal{T}$, with $G_{\text{target}}$ as the root. Reaction nodes follow an "AND" relation, requiring all child reactants, while molecule nodes follow an "OR" relation, meaning the product can be synthesized by any child reaction (Chen et al., 2020).

**Selection:** We select nodes from the frontier $\mathcal{F}(\mathcal{T})$ containing unexplored molecule nodes to expand the tree. Given an oracle cost function $J(\cdot)$, the next node is selected as $G_{\text{next}} = \arg\min_{G \in \mathcal{F}(\mathcal{T})} J(G)$ to minimize the cost. A well-designed $J(\cdot)$ improves search efficiency and aids in global optimality.

**Expansion:** After selecting $G_{\text{next}}$, a single GNN predictor call can generate many one-step retrosynthesis proposals. The GNN provides top-candidate reaction templates, each linked to different reactants. Thus we can form molecule nodes under the reaction node as an AND-OR stump.

**Update and Cost:** After expanding $G_{\text{next}}$, the tree becomes $\mathcal{T}'$. We update the nodes in $\mathcal{T}'$ for the next iteration. A* selects the path that minimizes $J(\cdot) = J_{\text{current}}(\cdot) + J_{\text{heuristic}}(\cdot)$, which includes the cost from the start to the current node $J_{\text{current}}(\cdot)$ and a heuristic estimate of the cost to the goal $J_{\text{heuristic}}(\cdot)$. With the GNN predictor, the negative log-likelihood of the reaction can be used to compute path cost $J_{\text{current}}(\cdot)$ to the leaf molecule node, we design $J_{\text{heuristic}}(\cdot)$ with the LLM in Llamole.

Figure 3: Overview of Llamole: Trigger tokens (`<design>` and `<retro>`) switch active modules from the base LLM to the respective graph component. The subsequent `<query>` token utilizes output vectors from the LLM to summarize past texts as conditions. Using these, Llamole generates molecules and predicts one-step reactions. Enhanced with a graph encoder and A* search, Llamole efficiently plans synthesis routes through selection and expansion iterations on the AND-OR Tree.

# 3 LLAMOLE: LARGE LANGUAGE MODEL FOR MOLECULAR DISCOVERY

## 3.1 MULTIMODAL AUTOREGRESSIVE MODELING

In molecular discovery, the sequence may include molecular structures $\mathcal{G}$ and retrosynthetic reactions $\mathcal{R}$ with each molecule or reaction tokenized. The sequence $Y = \{y_1, y_2, \ldots, y_N\}$, where $y_i \in \mathcal{W} \cup \mathcal{G} \cup \mathcal{R}$, combines these tokens. The sequence is interleaved with tokens in different spaces. Suppose the molecule appears at position $i$; then, we typically see:

$$\ldots, \quad Y_i \in \mathcal{G}, \quad Y_{i+1:i+L} \in \mathcal{W}, \quad Y_{i+L+1} \in \mathcal{R}, \quad \ldots$$

where $L$ is the length of the text following the molecule at position $i$. The sequence starts with text. If position $i$ denotes the first molecule in the sequence, then $Y_{<i} \in \mathcal{W}$; otherwise, $y_{i-1} \in \mathcal{R}$. To handle non-word tokens, we integrate domain-specific Graph DiT and GNN with the LLM, forming a multimodal LLM, i.e., Llamole. Parameterized by $\Theta$, Llamole unifies the cross-entropy losses from Eqs. (1) to (3) into autoregressive modeling:

$$\mathcal{L}_{\text{Llamole}} = \mathcal{L}_{\text{LM}} + \mathcal{L}_{\text{DM}} + \mathcal{L}_{\text{predictor}} = \sum_i -\log p_\Theta(y_i | Y_{<i}). \tag{4}$$

$\mathcal{L}_{\text{DM}}$ interprets $Y_{<i}$ as the input conditions, including desirable molecular properties and text conditions $\{c_i\}_{i=1}^M \cup \{\mathbf{c}_{\text{text}}\}$ for the autoregression of $Y_i$ in $\mathcal{G}$. In $\mathcal{L}_{\text{predictor}}$, $Y_{<i}$ represents $G_{\text{product}}$ and $\mathbf{c}_{\text{text}}$. Here, $G_{\text{product}}$ is generated from previous diffusion models or as intermediate $G \notin \mathcal{G}_{\text{avail}}$ in retrosynthesis. The autoregression for the label $Y_i$ is performed in the reaction space $\mathcal{R}$.

We present an overview of multimodal autoregression with Llamole in Figure 3, divided into controllable molecular generation and retrosynthetic planning. The base LLM performs multiple roles: generating text, controlling the switch of active modules, and providing cost functions for A* search. Augmented with the graph models, the overall parameters in Llamole are $\Theta = \{\theta_1, \theta_2, \theta_3, \phi_1, \phi_2, \phi_3\}$, where $\phi_1$ and $\phi_2$ project text into $\mathbf{c}_{\text{text}}$ for the Graph DiT and GNN predictor, respectively. The graph encoder with $\phi_3$ projects molecule tokens into the LLM. Next, we detail the design space of Llamole.

## 3.2 LLAMOLE DESIGN SPACE

Llamole consists of a base LLM and two pre-trained graph modules: the Graph DiT for molecule generation and the GNN for one-step reaction prediction. The base LLM employs a trigger-query-prediction approach using two sets of special tokens to switch between modules.

**Trigger Tokens.** Llamole defines two special trigger tokens to augment the word vocabulary $\mathcal{W}$: `<design>` for switching between the LLM and Graph DiT, and `<retro>` for switching between the LLM and GNN predictor. When a trigger token is predicted, Llamole activates the corresponding graph model. After molecule generation or reaction prediction, the active modules revert to the LLM.

**Query Tokens.** We introduce another set of special tokens, named query tokens `<query>` automatically placed after triggers. They use the LLM to query previous tokens and output hidden states as $\mathbf{c}_{\text{hidden}}$. A linear layer is applied: $\mathbf{c}_{\text{text}} = \text{Linear}(\mathbf{c}_{\text{hidden}})$, adjusting the input size for the graph models. We use different query tokens for different triggers. Query tokens allow us to share parameters $\phi_1$ and $\phi_2$ with $\theta_1$, enhancing both efficiency and effectiveness. We can apply ensemble methods by repeating the query tokens multiple times and averaging the $\mathbf{c}_{\text{hidden}}$ values (Dong et al., 2023).

Besides the special tokens, Llamole enhances molecule understanding with a graph encoder and uses the LLM to provide the cost function in A* search for retrosynthetic planning.

**Graph Encoder.** The graph encoder parameterized by $\phi_3$ replaces the word encoder in the LLM tokenizer for molecule tokens. The LLM decoder takes molecule embeddings from the graph encoder, along with text embeddings from the tokenizer, into the Transformer layers for next token generation. We use a pre-trained Graph Isomorphism Network (GIN) (Xu et al., 2018) as the graph encoder, optimized via molecule-text contrastive learning similar to CLIP (Radford et al., 2021).

**A* Cost Function with LLM.** We define $J_{\text{heuristic}}$ as a multi-choice problem, where each choice, assigned a score, represents synthesis complexity, from few to many steps. The LLM estimates the remaining synthesis steps for the leaf molecule node $G \in \mathcal{F}(\mathcal{T}) \setminus \mathcal{G}_{\text{avail}}$ in the search tree $\mathcal{T}$. It outputs probabilities for each choice, and $J_{\text{heuristic}}$ is computed as the weighted score by averaging the scores with their probabilities. For $G \in \mathcal{F}(\mathcal{T}) \cap \mathcal{G}_{\text{avail}}$, $J_{\text{heuristic}} = 0$.

### 3.3 END-TO-END MODEL FINE-TUNING AND GENERATION

**Supervised Fine-Tuning.** We use multimodal SFT to connect the base LLM and other graph modules in Llamole (Ouyang et al., 2022). Specifically, we freeze the parameters for the graph modules ($\theta_2$ and $\theta_3$) and fine-tune the LLM parameters $\theta_1$, the learnable special tokens, and the linear layers for the query tokens ($\phi_1$ and $\phi_2$). We freeze the parameters of the pre-trained graph encoder ($\phi_3$) and add a tunable linear layer between it and the LLM decoder. The optimization can be conducted end-to-end with Eq. (4). The SFT aligns the LLM with domain-specific graph models. To maintain generality in the base LLM, we employ parameter-efficient LoRA (Hu et al., 2021).

**Interleaved Generation.** Given a question as shown in Figure 2, Llamole performs controllable and synthesizable molecular designs, as presented in Figure 3. For the controllable generation, Llamole uses the base LLM to analyze the requirements and switches to the Graph DiT for generating $G_{\text{target}}$ when the trigger is predicted. For the synthesizable generation, Llamole plans synthesis routes for $G_{\text{target}}$. A* search on the AND-OR tree $\mathcal{T}$ aids in multi-step generation, interleaving molecule and reaction nodes, with $G_{\text{target}}$ as the root. During each selection-expansion iteration, A* selects $G_{\text{next}} = \arg\min_{G \in \mathcal{F}(\mathcal{T})} J(G)$ from the leaf nodes $\mathcal{F}(\mathcal{T})$. The graph encoder embeds molecule tokens into the LLM, which generates reaction conditions until the token `<retro>` is triggered, activating the GNN predictor. The predictor then predicts the top-50 templates as reaction nodes, along with corresponding reactants as molecule nodes for the next iteration. A* stops after finding a route from $G_{\text{target}}$ to $\mathcal{G}_{\text{avail}}$ with satisfying all AND-OR constraints, or if it fails after 30 seconds or 300 iterations. Upon success, the text with the corresponding reaction along the route is returned for retrosynthesis; otherwise, the base LLM directly generates texts.

## 4 BENCHMARKING FOR MULTIMODAL MOLECULAR DESIGN

To train Llamole, we need instruction data that provide detailed language supervision and evaluation covering synthetic complexity, drug and material utility, and reaction conditions. However, existing data based on PubChem (Kim et al., 2021) are only usable for small molecules and lack such details. Thus, we create MolQA, a large-scale graph-text multimodal instruction dataset for systematic LLM benchmarking used in Section 5. We also create MolPair with graph-text and reaction-text pairwise data to pre-train graph modules, as detailed in appendix D. To this end, we first collect multisource molecule data (Figure 4), with details in appendix C. Then we create MolQA and MolPair.

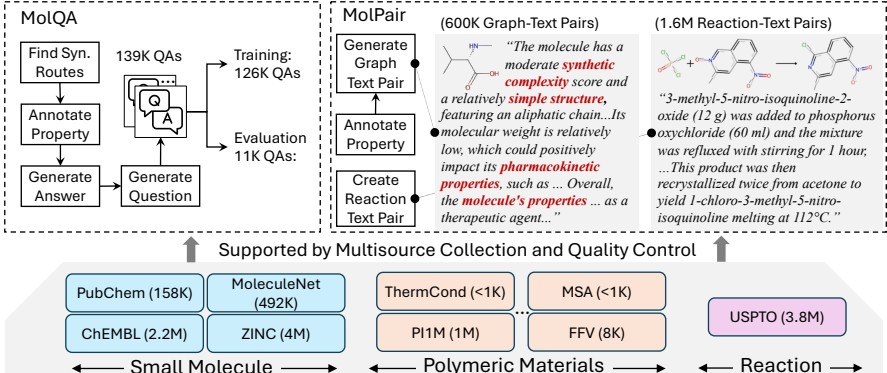

Figure 4: Creation of MolQA and MolPair: MolQA comprises two sets: a training set for ICL and (multimodal) SFT, and a test set for evaluation. MolPair consists of graph-text and reaction-text pairs, with red highlights indicating synthetic complexity, structure, and properties information.

**MolQA: Instruction Data Creation.** USPTO reactions include text descriptions. We use Enamine's 1.3 million small molecules as $\mathcal{G}_{\text{avail}}$. The depth-first search identifies routes from reaction products (i.e. target molecules) to molecules within $\mathcal{G}_{\text{avail}}$, resulting in about 139K routes with lengths ranging from 1 to 10. We sample around 11K routes (750 for materials and 9986 for drugs) for testing and use the rest for instruction tuning. We focus on eight popular properties for benchmarking (i.e, $M = 8$ for $\{c_i\}_1^M$ in Eq. (2)). They include three drug-related categorical properties (Wu et al., 2018): (1) HIV virus replication inhibition (HIV), (2) blood-brain barrier permeability (BBBP), and (3) human $\beta$-secretase 1 inhibition (BACE) and five continuous material properties (Thornton et al., 2012): (4) $CO_2$ permeability, (5) $N_2$ permeability, (6) $O_2$ permeability, (7) fractional free volume (FFV), and (8) thermal conductivity (TC). Not all target molecules have these properties. To enrich properties and texts, two supervised GNNs predict drug and material properties with confidence scores as in Liu et al. (2022; 2023a). Only high-confident predictions are selected for annotation. Llama-3-70B then generates descriptions using a template that incorporates these properties with structural and synthesis information from toolkits like RDKit. There are no polymerization reactions; we consider the monomer structure of the polymer as the synthesis target. We assemble molecule descriptions, text, and reactions from synthesis routes as answer data. Then Llama-3-70B is prompted to generate questions, resulting in MolQA with the example as shown in Figure 2. Details are in appendix C.2.

**MolPair: Pairwise Data Creation.** After excluding the target molecules from the instruction data, we use the remaining text-reaction data from USPTO to pre-train the GNN reaction predictor. Similarly, we utilize all other small molecules and polymers to pre-train the Graph DiT and graph encoder. For generalization, we expand beyond the eight properties used in the instruction data. For drug utility, we train another GNN to predict 41 properties, including toxicity, safety, enzyme interaction, absorption, distribution, metabolism, excretion (ADME), and biological activity (Swanson et al., 2024). For material utility, we consider 14 properties, such as thermal, physical, thermodynamic, permeability, solubility, and dielectric properties. Llama-3-70B generates related texts for these properties, incorporating structural and synthetic information. Finally, there are around 600K graph-text pairs for both small molecules and polymers to support pre-training. Details are in appendix C.3.

## 5 EXPERIMENT

We conduct a systematic evaluation to demonstrate Llamole's superior performance in controllable and synthesizable molecular design (RQ1). We investigate Llamole's performance in controllable molecular generation through ablation and case studies (RQ2). We analyze retrosynthetic performance of LLMs, focusing on error analysis and the efficiency and effectiveness of Llamole (RQ3).

**Set-ups:** We include LLM baselines from 7B to 70B, such as Llama, Mistral, Qwen, Granite, and Flan-T5, using either ICL or LoRA-based SFT. We also include domain-specific methods, GraphGA (Gao et al., 2022), DiGress (Vignac et al., 2022), and BioNavi (Zeng et al., 2024), for comparison. The MolQA test set contains 9,986 QA pairs for material design and 750 for drug design. LLMs are prompted with questions to generate responses for texts, molecules, and reactions. For controllability, we evaluate up to 12 metrics across four aspects: (1) chemical validity, (2)

Table 1: Multi-Conditional Molecular Design with LLMs: Best overall results in each metric are in **bold**, best baseline results are in *italic*. Balanced Accuracy (BA) = $\frac{\text{True Positive Rate}+\text{True Negative Rate}}{2}$.

| Base LLM or Method | Structure (↑) | | Text (↑) | | Drug (BA ↑) | | | Material (MAE ↓) | | | | |
|---|---|---|---|---|---|---|---|---|---|---|---|---|
| | Validity | Similarity | BLEU-4 | ROUGE-L | HIV | BBBP | BACE | $CO_2$Perm | $N_2$Perm | $O_2$Perm | FFV | TC |
| GraphGA | *0.885* | 0.112 | NA | NA | *0.536* | 0.515 | 0.560 | 0.847 | *1.556* | 0.747 | **0.020** | *0.042* |
| DiGress | 0.375 | 0.046 | NA | NA | 0.515 | *0.522* | *0.580* | *0.655* | 1.884 | *0.680* | **0.020** | 0.049 |
| **In-Context Learning** | | | | | | | | | | | | |
| Llama-2-7B | 0.167 | 0.024 | 0.030 | 0.141 | 0.051 | 0.060 | 0.053 | 5.463 | 3.982 | 4.943 | 0.308 | 0.199 |
| Mistral-7B | 0.251 | 0.044 | 0.066 | 0.203 | 0.163 | 0.153 | 0.200 | 5.062 | 3.824 | 4.657 | 0.289 | 0.186 |
| Qwen2-7B | 0.180 | 0.012 | 0.030 | 0.147 | 0.089 | 0.091 | 0.085 | 5.552 | 4.251 | 5.068 | 0.322 | 0.211 |
| Llama-3-8B | 0.656 | 0.112 | 0.155 | 0.307 | 0.471 | 0.473 | 0.562 | 3.233 | 3.106 | 2.924 | 0.171 | 0.123 |
| Flan-T5-XXL | 0.570 | 0.094 | *0.226* | *0.388* | 0.329 | 0.333 | 0.403 | 2.869 | 3.039 | 2.799 | 0.165 | 0.120 |
| Granite-13B | 0.498 | 0.079 | 0.170 | 0.326 | 0.260 | 0.293 | 0.285 | 2.994 | 3.165 | 2.993 | 0.180 | 0.123 |
| Llama-2-13B | 0.346 | 0.058 | 0.121 | 0.279 | 0.236 | 0.250 | 0.259 | 5.031 | 4.285 | 4.816 | 0.291 | 0.184 |
| Mistral-8x7B | 0.546 | 0.094 | 0.181 | 0.345 | 0.345 | 0.346 | 0.388 | 3.695 | 3.150 | 3.440 | 0.191 | 0.138 |
| Llama-2-70B | 0.299 | 0.045 | 0.099 | 0.222 | 0.237 | 0.242 | 0.274 | 5.368 | 4.336 | 5.017 | 0.319 | 0.202 |
| Llama-3-70B | 0.706 | 0.124 | 0.210 | 0.367 | 0.415 | 0.403 | 0.484 | 2.659 | 2.848 | 2.421 | 0.135 | 0.099 |
| **Supervised Fine-tuning** | | | | | | | | | | | | |
| Mistral-7B | 0.718 | 0.125 | 0.105 | 0.216 | 0.460 | 0.483 | 0.515 | 3.269 | 3.094 | 2.985 | 0.184 | 0.128 |
| Qwen2-7B | 0.768 | 0.133 | 0.221 | 0.377 | 0.436 | 0.457 | 0.457 | 2.691 | 2.562 | 2.721 | 0.147 | 0.106 |
| Llama-3-8B | 0.797 | *0.136* | 0.093 | 0.206 | 0.426 | 0.445 | 0.440 | 2.222 | 2.322 | 2.119 | 0.110 | 0.086 |
| Llama-3.1-8B | 0.692 | 0.121 | 0.121 | 0.250 | 0.417 | 0.432 | 0.433 | 3.210 | 2.991 | 2.974 | 0.179 | 0.122 |
| **Llamole** | | | | | | | | | | | | |
| Mistral-7B | 0.900 | 0.139 | **0.262** | **0.434** | 0.596 | 0.617 | 0.740 | **0.593** | 1.409 | 0.565 | 0.021 | 0.028 |
| Qwen2-7B | 0.888 | 0.135 | 0.261 | 0.432 | 0.600 | **0.639** | **0.746** | 0.645 | 1.452 | 0.581 | 0.021 | **0.026** |
| Llama-3.1-8B | **0.913** | **0.142** | 0.254 | 0.427 | **0.623** | 0.629 | 0.713 | 0.653 | **1.344** | **0.549** | 0.021 | 0.030 |
| **Improvement of Llamole (%)** | | | | | | | | | | | | |
| vs. All | +3.2 | +4.4 | +15.9 | +11.9 | +16.2 | +22.4 | +31.7 | +9.5 | +6.7 | +19.3 | -5.0 | +28.6 |
| vs. LLMs | +14.6 | +4.4 | +15.9 | +11.9 | +32.3 | +32.3 | +32.7 | +70.6 | +37.5 | +72.6 | +80.9 | +65.1 |

similarity to the reference based on Morgan fingerprints (Rogers & Hahn, 2010), (3) BLEU-4 and ROUGE-L scores against reference texts, and (4) deviation from desired properties. We follow Gao et al. (2022) to use well-trained random forests as the oracle functions for obtaining properties of designed molecules. We focus on three drug-related categorical properties assessed by balanced accuracy (BA) and five continuous material properties assessed by mean absolute error (MAE). For retrosynthesis, we evaluate the success rate of designed molecules against those available in $\mathcal{G}_{\text{avail}}$ from Enamine. Details are in appendix E.1.

## 5.1 RQ1: LLMs for Controllable and Synthesizable Molecular Design

Table 1 and Table 2 detail LLM performance in controllability and retrosynthesis success rate. The overall performance rankings are summarized in Figure 5. Our key observations are:

(1) **Llamole significantly outperforms other LLMs in text generation, controllable molecule generation, and retrosynthetic planning.** Llamole fine-tuned on various 7B-parameter LLMs, as shown in Table 2, results in top-3 rankings, surpassing 70B models that are 10× larger across all 12 metrics for controllability and planning success. Specifically, Llamole enhances chemical structure validity by 14.6%, structure controllability by 4.4%, and text generation by 11.9%-15.9%. Additionally, Llamole improves property controllability by 32% to 80%. In retrosynthesis, Table 2 indicates Llamole increases the success ratio from 5% to 35% for drugs and to 17.9% for polymers.

(2) **SFT improves molecular design but may not always enhance retrosynthesis.** According to Figure 5 and Table 1, SFT enables 7B LLMs to achieve chemical validity, structure, and property control comparable to 70B LLMs with ICL. However, it offers minimal improvement in planning ability for the generated target molecule. A notable example is Llama-3-8B from Table 2, where SFT reduces its retrosynthesis planning success from 5.5% to below 1%. Except for Llama-3-8B, we connect LLM performance with the same baseline but different learning methods in Figure 5. The results show that SFT methods still outperform ICL with the same base 7B models in most cases.

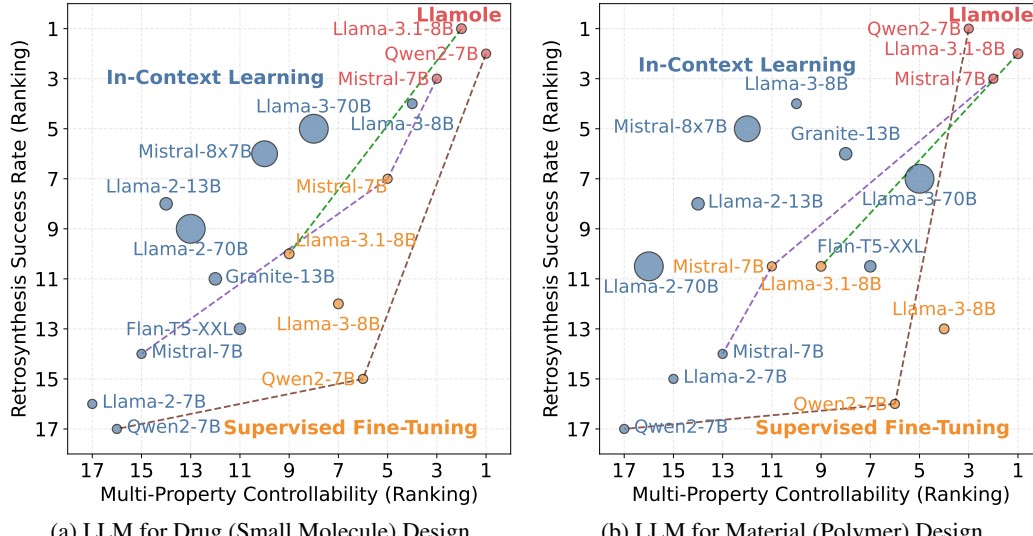

(a) LLM for Drug (Small Molecule) Design  (b) LLM for Material (Polymer) Design

Figure 5: Overall Comparison of LLMs for Controllability and Synthesizability: Performance is ranked by averaged BA/MAE (x-axis) and retrosynthesis success rate (y-axis). Circle size indicates model size. LLMs with ICL, SFT, and Llamole are highlighted in blue, orange, and red, respectively.

Table 2: Retrosynthetic Success Rate: Best results are in **bold** , best baseline results are in *italic* .

| | **In-Context Learning** | | | | | | | | |
|---|---|---|---|---|---|---|---|---|---|
| | Llama-2-7B | Mistral-7B | Qwen2-7B | Llama-3-8B | Flan-T5-XXL | Granite-13B | Llama-2-13B | Mistral-8x7B | Llama-2-70B |
| Drug (%) | 0.1 | 0.2 | 0.0 | 5.5 | 0.4 | 0.6 | 1.2 | 1.6 | 1.0 |
| Material (%) | 0.3 | 0.4 | 0.0 | 4.8 | 0.8 | 1.6 | 1.2 | 1.7 | 0.8 |

| | **Supervised Fine-tuning** | | | | **BioNavi for** | **Llamole** | | |
|---|---|---|---|---|---|---|---|---|
| | Mistral-7B | Qwen2-7B | Llama-3-8B | Llama-3.1-8B | DiGress | Mistral-7B | Qwen2-7B | Llama-3.1-8B |
| Drug (%) | 1.5 | 0.2 | 0.6 | 0.8 | *18.0* | 29.9 | 33.7 | **35.1** |
| Material (%) | 0.8 | 0.1 | 0.7 | 0.8 | *15.4* | 14.3 | **17.9** | 17.6 |

(3) **Larger models without domain-specific adaptation do not necessarily perform better in molecular designs.** We calculate the average Pearson correlation coefficient between model size and molecular design metrics, yielding a value of 0.366, indicating a weak correlation (below 0.5) between size and performance. We also compare LLM performance with GraphGA, which has been shown to be simple yet powerful (Gao et al., 2022; Liu et al., 2024c). Our observations confirm that GraphGA serves as a strong molecular design baseline, challenging most LLM models with ICL and SFT in generating molecules with precise multi-condition control.

## 5.2 RQ2: DISCUSSION ON CONTROLLABLE MOLECULAR GENERATION

### 5.2.1 ABLATION STUDIES ON LLM AND GRAPH DIT SYNERGY

We investigate the synergy effect of Graph DiT and LLM in Llamole for molecule controllability. We first remove text conditions $c_{text}$. In this case, Graph DiT uses a learned "null" embedding to represent the dropped condition $c_{text} = \emptyset$. Next, we remove the drug or material property conditions $\{c_i\}_i^M$ associated with the question. Results in Figure 6 show that text instructions enhance the chemical structure understanding ability of Graph DiT, while Llamole leverages Graph DiT's capabilities with property inputs to generate molecules with desirable properties.

### 5.2.2 CASE STUDIES FOR PROPERTY AND STRUCTURE CONTROLLABILITY

In Figure 7, Llamole can design a satisfactory molecule that meets both functional and structural constraints. Functionally, the oracle function confirms that the properties of BACE and HIV align with the criteria. Structurally, all key criteria are satisfied, including molecular weight, "two aromatic rings," and "connected to aliphatic chains." Llamole also adds details for structure design, such as a carboxyl ($-COOH$) group and an amino group ($-NH_2$). While the amino group is present in the

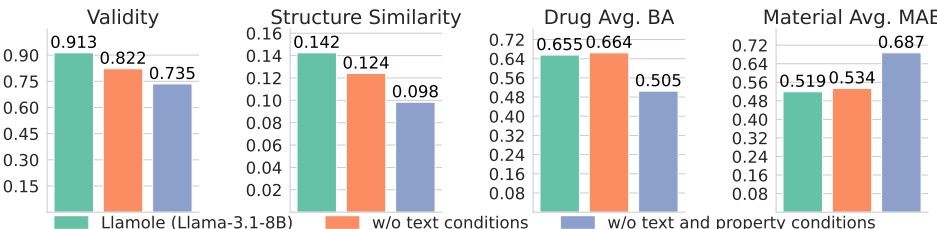

Figure 6: Ablation Studies for the Graph DiT Module in Llamole: First, we remove the text conditions from the input, i.e., $\mathbf{c}_{\text{text}} = \emptyset$. Next, we remove both text and property conditions, $\{c_i\}_i^M \cup \mathbf{c}_{\text{text}}$. There are learned embeddings that represent the "null" value for different conditions.

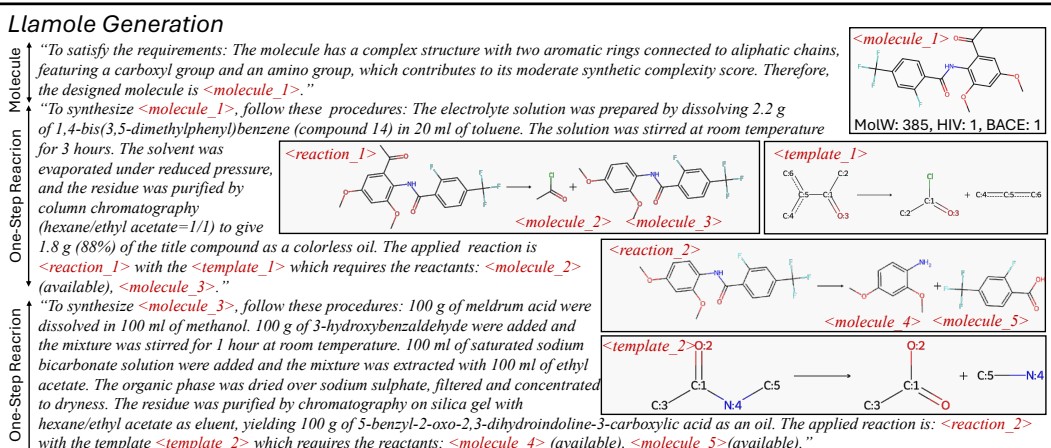

Figure 7: Interleaved generation with the base Qwen2-7B: Red indicates positions where molecules and reactions (with templates) are generated, forming three parts. The properties of the designed molecules are obtained from the oracle. Reference and other LLM responses are shown in Figure 9.

structure, it is connected to the carbonyl group $(-C(=O)-)$ instead of the carboxyl group. This subtle difference may require precise control based on the text condition. More results are in appendix E.3.

## 5.3 RQ3: Discussion on Retrosynthetic Planning

Retrosynthesis challenges LLMs in two aspects: (1) one-step reaction generation and (2) multi-step planning. Table 2 highlights the weaknesses of LLMs with ICL and SFT in overall planning ability and the promise of Llamole. We examine the failure reasons in LLMs and the synergy between the GNN and LLMs to avoid them.

### 5.3.1 One-step Reaction Generation

We conduct error analysis for LLMs in reaction generation. Results in Figure 8 average performance across all LLMs using ICL or SFT methods. We identify five types of errors related to instruction adherence, format compliance, and template matching. We find that LLMs using ICL frequently fail to follow instructions for generating reactions in text format, with a high probability (68.4%) of not producing valid formats and templates. In contrast, LLMs with SFT reduce this probability to 57.6%. However, neither ICL nor SFT guarantees that the templates are correct or match the generated reactions. In comparison, Llamole avoids these errors by using GNN predictors, which estimate probabilities for over 300K templates derived from USPTO reactions. This enables Llamole to apply templates directly to derive reactions in retrosynthesis, avoiding hallucination.

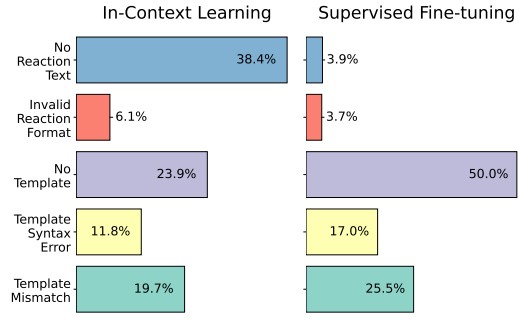

Figure 8: Error Analysis in Reaction Generation

### 5.3.2 Multi-step Retrosynthetic Planning

From the success cases in Table 2, we find that 96.40% of 777 success cases in ICL-adapted LLMs and 94.14% of 324 success cases in SFT-adapted LLMs arise from one-step reaction generation. However, not all designed molecules can be synthesized via one-step reactions. Compared to LLMs, Llamole achieves over 10K success cases, with 40.48% resulting from two or more steps. Figure 7 illustrates a two-step planning case for the designed molecule. The generation interleaves reaction conditions and specific formulas based on the template in both steps.

Llamole is influenced by two factors for retrosynthesis: (1) the size of the search space and (2) the quality of the cost $J_{heuristic}$. The results reported in Table 2 limited the total planning time to 30 seconds (on an A6000 card). We remove this time constraint and report comparisons for material tasks in Table 3. We find that success rates for all base LLMs significantly improve, but this comes at the cost of long inference time. While there is a trade-off between efficiency

Table 3: Analysis of $J_{heuristics}$ and Planning Time on Material Questions

| Base LLM | Default | w/ Domain Heuristics | w/ Unlimited Time |
|---|---|---|---|
| Llama-3.1 | 0.176 | 0.176 | 0.312 |
| Mistral | 0.143 | 0.147 | 0.273 |
| Qwen2 | 0.179 | 0.181 | 0.273 |

and effectiveness, it is often acceptable to extend response time by a few minutes to enhance success rates for finding synthesis paths. In Table 3, we also compare the $J_{heuristics}$ designed by LLMs (default) with the domain model trained from Chen et al. (2020). we find that LLMs are competitive with these domain models in providing the cost function for A*, contrasting with previous observations where LLMs struggled with retrosynthetic planning.

## 6 Related Work

Since the emergence of ChatGPT (Achiam et al., 2023), LLMs (Dubey et al., 2024) have become foundation models for text-based problems and are revolutionizing domains like vision and speech (Dong et al., 2023; Wu et al., 2024). These advancements extend to chemistry, biology, and material sciences, focusing on molecules (Guo et al., 2023; Jin et al., 2023). Prior work explores LLMs in molecular generation, property prediction, and one-step reaction prediction in retrosynthesis (Guo et al., 2023; Jablonka et al., 2023). A key lesson is the limitation of LLMs in sequential modeling of molecules (e.g., SMILES or SELFIES) (Guo et al., 2023). Multimodal LMs have been developed for molecular tasks (Edwards et al., 2022; Liu et al., 2023b), but they either do not treat molecules as graphs (Edwards et al., 2022) or do not focus on inverse molecular design. Additionally, LLMs struggle with planning tasks (Kambhampati et al., 2024), which are essential for retrosynthesis. We address these issues using graph-text multimodal LLMs, augmented by A* for efficient planning.

Domain-specific molecular design methods have evolved from sequential models (Segler et al., 2018) to graph diffusion models (Vignac et al., 2022; Weiss et al., 2023; Liu et al., 2024c). Studies show that older graph methods like GraphGA remain competitive (Gao et al., 2022). To incorporate property constraints, one can use Bayesian optimization or REINFORCE (Gao et al., 2022), or employ diffusion models with or without predictor guidance (Vignac et al., 2022; Liu et al., 2024c). For synthesizable molecular design, prior work has focused on bottom-up methods (Gao et al., 2021; Sun et al., 2024). These methods explore a chemical space defined by a discrete action space of reaction templates and purchasable starting materials, which may limit flexibility. Thus, retrosynthesis algorithms (Chen et al., 2020; Han et al., 2022; Zeng et al., 2024) are also studied as separate solutions to find synthesis routes for generated molecules in a top-down manner.

## 7 Conclusion

We have presented the first graph-text MLLM, Llamole, for multi-conditional molecular generation and retrosynthetic planning. By integrating a base LLM with specialized graph modules, Llamole interleaved the generation of text, molecular graphs, and reactions, enabling controllable and synthesizable designs. Extensive benchmarking against 14 LLMs revealed their limitations in controlling molecular structures and planning synthesis routes. In contrast, Llamole significantly outperformed these LLMs. These findings underscored the value of multimodal approaches in molecular discovery and highlighted Llamole's potential to connect text and chemical structures. The new benchmarking dataset also laid the groundwork for future MLLM research in molecular applications.

ACKNOWLEDGMENTS

This work was in part supported by NSF IIS-2142827, IIS-2146761, IIS-2234058, CBET-2332270, and ONR N00014-22-1-2507.

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

CONTENTS

## A   MORE RELATED WORK ON MULTIMODAL LANGUAGE MODELING

Emerging approaches focus on multimodal graph and language modeling for tasks such as molecular property prediction (Zhao et al., 2023), captioning (Edwards et al., 2022; Liu et al., 2024d), and retrieval (Liu et al., 2023b; 2024d). The task most similar to inverse molecular design is text-based molecular generation (Edwards et al., 2022; Liu et al., 2024d). In this work, inverse molecular design is framed as a question with specific requirements for properties and synthesis paths. Unlike text-based generation, which takes descriptions of molecules as input, inverse molecular design requires fewer details on the molecule, focusing instead on satisfying the specified requirements. Additionally, text-based generation produces molecular structures without considering synthesizability, whereas designed molecules are often expected to be synthesizable (Gao et al., 2021), involving retrosynthesis.

Table 4: Balanced Accuracy Averaged Across Three Drug Design Properties

| MolT5-small | MolT5-base | MolT5-large | Best LLM with ICL | Llamole |
|---|---|---|---|---|
| 0.150 | 0.232 | 0.264 | 0.502 | 0.662 |

To explore the difference between inverse molecular design and text-based generation, we use the decoder model from (Edwards et al., 2022; Liu et al., 2024d) (i.e., MolT5) and test questions from the MolQA benchmark to compare the performance of MolT5, LLMs, and Llamole in drug design. Results on balanced accuracy are shown in Table 4. We find that even the largest MolT5 underperforms the best LLM (from ICL) in drug design. This illustrates that text-based molecular generation, which takes descriptions of molecules as input, may not perform well in inverse molecular design, which requires satisfying specific properties with synthesis paths and a lack of details about the molecule in texts. For material design, we find that MolT5 cannot generate valid polymer structures due to its lack of knowledge about polymerization points, typically represented by the asterisk symbol in SMILES strings. As a result, no valid MAE error is reported. Additionally, existing multimodal language models have not addressed the retrosynthetic planning problem.

## B   ADDITIONAL DETAILS FOR LLAMOLE

### B.1   DETAILS OF SPECIAL TOKENS

In total, there are nine special tokens divided into three groups. These tokens augment the word vocabulary $\mathcal{W}$, enabling flexible control of the generation flow:

- Trigger and Query tokens: `<design_start>`, `<design_body>`, `<design_end>`, `<retro_start>`, `<retro_body>`, `<retro_end>`
- Molecule token: `<molecule>`
- Callback tokens: `<callback_start>`, `<callback_end>`

The tokens `<design_start>` and `<retro_start>` switch between the LLM and the Graph DiT or GNN, respectively. The tokens `<design_body>` and `<retro_body>` serve as query tokens, repeated eight times. After tokenization, the LLM takes their embeddings as input and outputs a vector from the last layer. The tokens `<design_end>` and `<retro_end>` indicate the end of these switches.

The `<molecule>` token marks the position of the molecular graph where the graph encoder is applied. In the instruction dataset, the segment "`<mol_start>`SMILES`<mol_end>`" denotes the position and identity of the molecule. SMILES will be converted to molecular graphs using RDKit, and this segment will be replaced by the `<molecule>` token for Llamole inputs.

Finally, callback tokens control the LLM to generate backup results as complements to the specialized graph modules. For instance, if the Graph DiT fails to produce a valid molecule, the base LLM can generate an alternative, regardless of validity.

### B.2 Details of LLM-based A* Heuristics

Llamole models $J_{\text{heuristics}}$ in A* search as a multi-choice problem, filling in information from the molecule node, its parent reaction nodes and siblings using the template below. Parameters such as step, reaction template, and reactants are optional.

```
Estimate remaining steps for the target {smiles} given the
following parameters:
Current step {step},
Current template: {template},
Reactants: {reactants}.
Consider the following factors:
1. Intermediate complexity
2. Reagent availability
3. Side reactions
4. Stereochemistry challenges.
```

Using this question to estimate remaining steps, we input the text into the base LLM and formulate five choices with corresponding scores:

```
A. All readily available // Score: 0
B. Some commercial, some need 1-2 steps // Score: 1
C. Mix of commercial and multi-step synthesis // Score: 2.5
D. Mostly require complex synthesis // Score: 4.5
E. All require extensive multi-step synthesis // Score: 7
```

The LLM outputs logits for the next token, which we average for each choice to obtain overall probabilities. The $J_{\text{heuristics}}$ is calculated as the weighted score using these probabilities.

## C  Additional Benchmarking and Datasets Details

We collect small drug molecules from PubChem (Kim et al., 2021), MoleculeNet (Wu et al., 2018), ChEMBL (Zdrazil et al., 2024), and ZINC (Sterling & Irwin, 2015). Polymers are macromolecules with one repeating unit called monomers. We collect polymers from PI1M (Ma & Luo, 2020), the Membrane Society of Australia (MSA) (Thornton et al., 2012), and others (Liu et al., 2024b). Additionally, we collect 3.8 million patent chemical reactions with descriptions from USPTO (Lowe, 2017), spanning from 1976 to 2016.

### C.1 Details of Quality Control

After collecting molecules and polymers from various sources, we deduplicate and merge the label information for identical molecules. We use RDKit to obtain canonical SMILES. For small molecules, we calculate the first 14 characters of the InChIKey as the unique identifier, while for polymers, where the polymerization point is represented by "*", we use the canonical SMILES directly.

For drug-like small molecules, we apply the following rules to filter out alert structures, known as the Rule of Five (Ro5):

- Molecular Weight (MW): Must be $\leq 500$ Da.
- Hydrogen Bond Acceptors (HBA): Must not exceed 10.
- Hydrogen Bond Donors (HBD): Must not exceed 5.
- LogP: Must be $\leq 5$, indicating lipophilicity.

A molecule passes the Ro5 test if at least three of these four conditions are met, indicating potential oral bioavailability.

We also apply 15 filter rules from the RDKit package, including the following from the Fil-terCatalogs Class: BRENK, CHEMBL, CHEMBL_BMS, CHEMBL_Dundee, CHEMBL_Glaxo, CHEMBL_Inpharmatica, CHEMBL_LINT, CHEMBL_MLSMR, CHEMBL_SureChEMBL, NIH, PAINS, PAINS_A, PAINS_B, PAINS_C, and ZINC.

## C.2 Details on the Creation of MolQA

### C.2.1 Creation of Synthesis Routes

The USPTO has 3.7 million reactions. There are approximately 1.3 million unique product molecules. The purchasable compounds come from the Enamine Building Block (June 2024 version), supplemented with other common ions and starting materials, totaling around 1.3 million. We check each product from USPTO as a target molecule in the retrosynthesis task, exploring whether they can be synthesized using existing USPTO reactions through depth-first search (DFS). Ultimately, we identify about 139K target molecules with synthesis routes, supporting the creation of MolQA.

Since there are no polymerization reactions, we consider only monomer structures by replacing the * point with hydrogen. Among the 139K small molecules with synthesis routes, 2196 fit the monomer structures and serve as target molecules for polymer retrosynthesis. The length of synthesis routes ranges from 1 to 10. For each length of the routes, we split half of the molecules into the testing set, with a maximum of 3000, while the remainder is retained in the training set.

It results in around 11K routes (750 for materials and 9986 for drugs) for testing and 126K target molecules for training.

### C.2.2 Creation of Property Annotations

We focus on eight benchmarking properties: three drug-related categorical properties (Wu et al., 2018)—(1) HIV virus replication inhibition (HIV), (2) blood-brain barrier permeability (BBBP), and (3) human $\beta$-secretase 1 inhibition (BACE)—and five continuous material properties (Thornton et al., 2012)—(4) $CO_2$ permeability ($CO_2$Perm), (5) $N_2$ permeability ($N_2$Perm), (6) $O_2$ permeability ($O_2$Perm), (7) fractional free volume (FFV), and (8) thermal conductivity (TC).

First, we check existing sources for annotations of these properties. To enrich the label space, we use well-trained GNN models (Liu et al., 2022) to generate confident pseudo-labels, following the method in (Liu et al., 2023a). We collect all labeled data to train two supervised multi-task GIN models for drug and material property annotation. The GIN models employ rationalization techniques (Liu et al., 2024a) to split the molecular graph into rationale and environment subgraphs in the latent space, predicting labels from the rationale subgraph. The confidence score is computed by combining the rationale subgraph with various environment subgraphs, using the reciprocal of prediction variance. We annotate properties when prediction confidence exceeds the median threshold.

### C.2.3 Creation of Text Data for Molecular Description

In addition to property annotations, we consider structural and synthesis information of the molecules using RDKit and heuristic complexity estimation scores. First, for any molecule, we extract the following structural information:

- **Scaffold:** Extracted scaffold from the molecule structure.
- **Molecular Weight:** Calculated using the molecular weight descriptor.
- **Number of Rings:** Total number of rings in the molecule.
- **Number of Aromatic Rings:** Total number of aromatic rings in the molecule.
- **Number of Aliphatic Rings:** Total number of aliphatic rings in the molecule.
- **Number of Rotatable Bonds:** Total number of rotatable bonds in the molecule.
- **Number of Hydrogen Bond Donors:** Total number of hydrogen bond donors.
- **Number of Hydrogen Bond Acceptors:** Total number of hydrogen bond acceptors.

Next, we compute the synthetic accessibility score (SAScore) (Ertl & Schuffenhauer, 2009) and SCScore (Coley et al., 2018). Based on this information, we use the following template:

```
Generate a summary description that starts directly with "The
molecule/polymer ..." based on the predicted chemical properties,
synthetic complexity scores, and structural information for the
molecule with SMILES: {{smiles}}. Use your own knowledge, focus
```

```
on functions, and avoid using numbers, redundant words, or
mentioning SMILES. Ensure the output sentence is complete and
ends with a period. This is for Drug/Material Utility of a
Molecule/Polymer:

The structural context of a molecule includes its scaffold, which
is the core structure around which the molecule is built. Key
structural features include the presence of aromatic rings,
aliphatic chains, and common functional groups such as hydroxyl,
carboxyl, and amino groups. The complexity of the molecule's
structure can significantly influence its physical and chemical
properties.
Scaffold: {{scaffold}}
Molecular Weight: {{mw}}
Number of Rings: {{num_rings}}
Number of Aromatic Rings: {{num_arom_rings}}
Number of Aliphatic Rings: {{num_aliph_rings}}
Number of Rotatable Bonds: {{num_rot_bonds}}
Number of Hydrogen Bond Donors: {{num_h_donors}}
Number of Hydrogen Bond Acceptors: {{num_h_acceptors}}

{utility_context}
{{properties}}
```

The pre-defined utility context for the small molecule is as follows:

```
The drug utility of a molecule is assessed based on its potential
to serve as a therapeutic agent. Key properties considered
include pharmacokinetics, which encompasses absorption,
distribution, metabolism, excretion (ADME), and toxicity.
Bioactivity is another critical factor, measured by the
molecule's ability to interact with biological targets, typically
through binding affinity. Additionally, drug-likeness, which
refers to the molecule's adherence to established rules such as
Lipinski's Rule of Five, is essential. This rule evaluates
molecular weight, hydrogen bond donors and acceptors, and
lipophilicity to predict a molecule's suitability as an oral drug.
```

The pre-defined utility context for the polymer is as follows:

```
The material utility of a molecule, particularly for creating
polymeric materials, is evaluated based on properties like
mechanical strength, flexibility, and thermal and electrical
behavior. For polymer membranes used in gas separation, crucial
factors include gas permeability, which determines the efficiency
of gas diffusion, and chemical stability, ensuring resistance to
degradation. Additionally, thermal properties such as melting
point and thermal conductivity are vital, as they affect the
material's performance under various temperature conditions.
Electrical properties, such as conductivity and dielectric
constant, may also be significant depending on the intended
application.
```

For the property variable, we include the property name with values, as well as the minimum, maximum, and percentile among the labels in the template. We repeat all annotated properties in the property variable. The estimated synthesis complexity scores are included among them.

We also prompt Llama-3-70B to generate short responses of 50-70 words, producing a molecular description for each molecule based on its properties, structures, and synthesis estimation. If a molecule has a description from PubChem (Kim et al., 2021), we concatenate these descriptions.

The generated texts may not always be meaningful or valid. We can establish filter rules based on patterns observed in poorly generated texts to remove them. We then regenerate texts for these items. After several iterations, we obtain the final text data for molecular utility descriptions, improving overall text quality. We also apply this strategy to other steps that involves prompting LLMs for synthetic data creation.

### C.2.4 CREATION OF QUESTION ANSWERING DATA

After annotating molecular description texts from appendix C.2.3, we combine them with reaction descriptions, including the reaction formula and template from synthesis routes in appendix C.2.1. This forms the answer data in a QA data pair.

Next, we prompt Llame-3-70B to generate questions for each answer based on the following template.

```
I'm creating a question-answer dataset for LLM fine-tuning.
The question is about designing a molecule/polymer with these
properties: {property_info} and the following structure
information: {structure_info}.
The expected answer for the question is: {answer}
Generate a SINGLE question about designing and synthesizing such
a molecule/polymer that meets these criteria:
(1) Start with 'Question:'; (2) End with a question mark;
(3) Sound natural; (4) Be diverse; (5) Avoid redundancy and
introductory words (like 'Here is a question that meets the
criteria:')
(6) Do not include the answer; (7) Do not include incorrect
information.

Example questions:
(1) How can I design and synthesize a molecule with X, Y, and Z
properties?
(2) What is the best way to create a polymer with X, Y, and Z
characteristics?
(3) How to design a molecule with X, Y, and Z features and
synthesize it?
(4) I want a molecule with X, Y properties and Z structures.
Please design it and describe the synthesis path.
```

The template is applied to any answer with the corresponding structure, property information, and complete answer texts.

### C.3 DETAILS ON THE CREATION OF MOLPAIR

MolPair consists of two parts: reaction-text pairs and graph-text pairs. We curate reaction-text pairs from USPTO (Lowe, 2017), pairing each reaction with its corresponding description of the reaction conditions. We first deduplicate product molecules in reactions, obtaining input data as the product molecule alongside the reaction condition texts. Next, we extract reaction templates from the reaction formula using rdchiral (Coley et al., 2019), resulting in approximately 300K templates, which will serve as labels for predictions. Finally, we have approximately 1.6 million training examples.

For the graph-text pairs, we use small molecules and polymers from the multisource collection, excluding those in MolQA. We follow the same pipeline used to create property and text annotations for the MolQA data, focusing on broader properties that describe drug-related utility with 41 small molecule properties (Swanson et al., 2024). Besides the three used in MolQA, others include:

- Toxicity and Safety: AMES, Carcinogens Lagunin, ClinTox, DILI, Skin Reaction, hERG
- Enzyme Interaction: CYP1A2 Veith, CYP2C19 Veith, CYP2C9 Substrate CarbonMangels, CYP2C9 Veith, CYP2D6 Substrate CarbonMangels, CYP2D6 Veith, CYP3A4 Substrate CarbonMangels, CYP3A4 Veith

- Absorption, Distribution, Metabolism, and Excretion (ADME): BBB Martins, Bioavailability Ma, Caco2 Wang, Clearance Hepatocyte AZ, Clearance Microsome AZ, HIA Hou, Half Life Obach, Hydration Free Energy FreeSolv, Lipophilicity AstraZeneca, PAMPA NCATS, PPBR AZ, Pgp Broccatelli, Solubility AqSolDB, VDss Lombardo

- Stress Response: SR-ARE, SR-ATAD5, SR-HSE, SR-MMP, SR-p53

- Nuclear Receptor Interaction: NR-AR-LBD, NR-AR, NR-AhR, NR-Aromatase, NR-ER-LBD, NR-ER, NR-PPAR-gamma

We describe polymeric material utility based on 14 polymer properties collected from Otsuka et al. (2011):

- Thermal Properties: Melting temperature [°C]; Specific heat capacity at constant pressure ($C_p$) [cal/(g·°C)]; Specific heat capacity at constant volume ($C_v$) [cal/(g·°C)]; Thermal conductivity [W/(m·K)]

- Physical & Thermodynamic Properties: Density [g/cm$^3$]; Fractional Free Volume (dimensionless); Radius of Gyration ($R_g$) [nm]

- Permeability Properties: Gas diffusion coefficient ($D$) [cm$^2$/s]; Gas permeability coefficient ($P$) [cm$^3$ (STP)·cm/(cm$^2$·s·Pa)]; Oxygen ($O_2$) Gas Permeability (Barrer); Nitrogen ($N_2$) Gas Permeability (Barrer); Carbon Dioxide ($CO_2$) Gas Permeability (Barrer)

- Solubility Properties: Gas solubility coefficient ($S$) [cm$^3$ (STP)·cm/(cm$^2$·s·Pa)]

- Dielectric & Optical Properties: Dielectric constant.

We train two multi-task GIN models based on the rationalization method (Liu et al., 2022) using all existing labeled data for drug and material property prediction, respectively. We use these models to predict properties for millions of small molecules and polymers, retaining the top ten thousand predictions by confidence score for each property. These are then used to prompt Llama-3-70B to create molecular descriptions, using the same prompt template as in appendix C.2.3. Additionally, we apply the same strategy as in appendix C.2.3 to annotate labels for the eight studied properties, which can serve as input for pretraining the multi-conditional Graph DiT. Finally, we have approximately 300K graph-text pairs for small molecules and 300K graph-text pairs for polymers.

# D ADDITIONAL PRE-TRAINING AND FINE-TUNING DETAILS

We pre-train three graph models including Graph DiT (Liu et al., 2024c) for multi-conditional molecular generation, a GIN-based GNN predictor for reaction template prediction, and a GIN-based graph encoder for molecule understanding (Xu et al., 2018).

## D.1 PRE-TRAINING OF GRAPH DIFFUSION TRANSFORMER

Suppose the node has $F_V$ categories and the edge has $F_E$ categories (including non-bond). Graph DiT models the node token by concatenating all its edge configurations to other nodes. For each node $\mathbf{x} \in \mathbb{R}^F$, we have $F = F_V + N_G \times F_E$, where $N_G$ denotes the graph size. This facilitates defining the transition matrix $\mathbf{Q}$ for the joint distribution of nodes and edges (Liu et al., 2024c). Graph DiT uses Transformer layers, replacing layer normalization with adaptive layer normalization (AdaLN):

$$\text{AdaLN}\,(\mathbf{h}, \mathbf{c}) = \gamma_\theta(\mathbf{c}) \odot \frac{\mathbf{h} - \mu\,(\mathbf{h})}{\sigma\,(\mathbf{h})} + \beta_\theta(\mathbf{c}),$$

where $\mathbf{h}$ denotes the hidden state of $\mathbf{x}$ and $\mathbf{c}$ is the vector representing the input conditions.

Given multiple conditions with categorical, continuous properties, and text, Graph DiT uses one-hot encoding for categorical properties and a clustering-based approach with $\text{Linear}\,(\text{Softmax}\,(\text{Linear}(c)))$ to embed continuous condition values $c$. We employ pre-trained SciBERT (Beltagy et al., 2019) to embed input texts into a 768-dimensional vector by averaging the representations of all text tokens in the sentence, then using a linear layer to adjust the dimension for Graph DiT. For each condition, the model also learns a drop embedding. The drop embedding is used when no values are provided. Finally, the model sums the representation vectors of different

conditions as input for **c**. In the reverse diffusion process, the denoising model uses predictor-free guidance to sample molecular graphs given multiple conditions. We pre-train the denoising model with the loss function in Eq. (2) using 600K graph-text pairwise data and the eight properties defined in appendix C.3. The model employs the following hyperparameters: depth of 28, hidden size of 1024, 16 heads, and MLP hidden size of 4096. The total model size is around 574 million parameters. We pre-train the model for 45 epochs, which takes approximately one week on a single A100 card.

## D.2 Pre-training of GNNs

We pre-train a three-layer GIN to predict reaction templates among 30,124 labels, using a hidden size of 512. Reaction template prediction is a multi-class classification task. Given reaction-text pairs from MolPair, we extract the product molecular graph from the reaction formula, using the reaction condition text as input. SciBERT (Beltagy et al., 2019) is used as the text encoder with frozen parameters. We average the text representations to obtain a sentence-level representation. The prediction target is the reaction template extracted from the reaction (Coley et al., 2019). GIN naturally uses molecular graphs, employing the AdaLN approach as the normalization layer added after each message-passing layer to incorporate text conditions. We pre-train the model for 5 epochs on a single V100 card, with 632 million parameters. This model serves as the reaction predictor to suggest reaction templates for Llamole.

For molecular understanding, we pre-train a five-layer GIN model with a hidden size of 768. SciB-ERT (Beltagy et al., 2019) is used as the text encoder with frozen parameters. We average the text representations to obtain a sentence-level representation, while the GIN model uses sum pooling to produce the graph representation. For each graph-text pair from MolPair, we optimize the graph encoder using the CLIP loss (Radford et al., 2021) for 40 epochs. The CLIP loss consists of two contrastive losses: it first computes the similarity score between graph-text pairs, then contrasts it with all other similarity scores by pairing the graph with other texts and pairing the text with other graphs as negative pairs. The model has around 43 million parameters. The model can be pre-trained on a single V100 card in a few days. This graph encoder will replace the word encoder in the LLM tokenizer module for molecules indicated by the token `<molecule>` as shown in appendix B.

## D.3 Fine-tuning of Llamole

Llamole is fine-tuned on graph-text multimodal instruction data, freezing the parameters of the Graph DiT, GNN predictor, and graph encoder. It automatically adds eight query tokens to the sequence once the trigger tokens are predicted, allowing the base LLM to continue autoregression and output vectors for all eight query tokens. We average these output vectors as queries for prior generated texts and use them as input text vectors for the subsequent Graph DiT or GNN predictor module via a tunable linear layer. For the `<molecule>` token, we add a tunable linear layer on top of the token embedding after the graph encoder outputs it. Without loss of generality, we study three variants of Llamole with different base LLMs: Llama-3.1-8B (Dubey et al., 2024), Mistral-7B (Jiang et al., 2023), and Qwen2-7B (Yang et al., 2024). All LLMs are fine-tuned using LoRA (Hu et al., 2021) for four epochs, taking approximately two days on a single A100 card.

# E Additional Experimental Details and Discussions

## E.1 Additional Details on Experimental Set-ups

In Tables 1 and 2 and Figures 1, 5a and 5b, Llamole is compared with fourteen LLMs with sizes ranging from 7B to 70B, including Llama (Dubey et al., 2024), Mistral (Jiang et al., 2023), Qwen (Yang et al., 2024), Granite (Abdelaziz et al., 2024), and Flan-T5 (Chung et al., 2024). We prefer the instruct version of the model when available.

Using the MolQA training set, previous work can implement these LLMs in two ways: in-context learning (ICL) and text-only supervised fine-tuning (SFT). For ICL, we retrieve five closest QA pairs from the training set based on the average property difference from desired properties. The template used to construct the prompt with demonstrations is:

```
I'm working on designing and synthesizing molecules. Here are
some example questions and answers about molecular requirements,
design, and synthesis: {{examples}}
Now, based on these examples, please answer the following
question about molecular design and synthesis: {{question}}
```

For SFT, we fine-tune the LLMs with LoRA after converting molecules into SMILES strings.

The MolQA test set contains 9,986 QA pairs for small molecules in drug applications and 750 pairs for polymeric materials. The questions serve as input to prompt the LLMs to generate responses.

For the controllability of multi-conditional molecular generation, we evaluate up to 12 metrics across four aspects: (1) chemical validity, (2) similarity to the truth based on Morgan fingerprints, (3) BLEU-4 and ROUGE-L scores compared to reference texts, and (4) deviation from desired properties. For polymer validity, we further examine whether the generated molecular structures contain at least two polymerization points ("*"). To obtain the properties of the designed structure, we define an oracle function based on well-trained random forests from all annotated molecules, following previous work (Gao et al., 2022; Liu et al., 2024c). We evaluate three drug-related categorical properties using balanced accuracy (BA) and five continuous material properties using mean absolute error (MAE). As a baseline, we consider GraphGA (Gao et al., 2022) to reference the performance of LLMs compared to domain-specific methods.

For retrosynthesis, we evaluate the success rate from the designed molecule to those available in $\mathcal{G}_{\text{avail}}$, purchasable from the Enamine Building Block (June 2024 version), supplemented with other common ions and starting materials, totaling around 1.3 million.

### E.1.1 SET-UPS FOR FIGURE 1

For Figure 1, we average the balanced accuracy for three drug-related properties and five MAEs for the polymeric material properties. We then select the model with the best performance in each category based on these average metrics. For drug tasks, the best ICL model is Llama-3-8B-ICL, the best SFT model is Mistral-7B-SFT, and the best Llamole variant is based on Qwen2-7B. For material tasks, the best ICL model is Llama-3-70B-ICL, the best SFT model is Llama-3-8B-SFT, and the best Llamole variant is based on Llama-3.1-8B. Their average performance is visualized in Figure 1 in comparison with GraphGA.

### E.1.2 EXTRACTION OF SMILES FROM LLM RESPONSES

ICL or SFT-based LLMs generate free-form text that includes both natural language and SMILES-represented molecular structures. We need a method to automatically extract SMILES strings from LLM outputs for evaluation. Practically, one can observe generation patterns to summarize rules for regular expressions to accomplish this. In the MolQA training set, the designed molecular structures typically follow the phrase "the designed molecule is:" as shown in examples Figures 9 and 10. LLMs may not always adhere strictly to this pattern, so we may need to extend this rule to cover more cases. In the future, more sophisticated regular expressions could be developed to extract SMILES strings from text directly. However, these will still need to be combined with additional rules to identify the designed molecules, as LLMs may generate intermediate SMILES strings before and after the designed molecule. Compared to them, Llamole uses `<design_start>` or `<retro_start>` to indicate the position of generated molecular structures.

### E.2 ADDITIONAL DISCUSSION ON ONE-STEP GENERATION

We further examine the text generation results for reaction conditions. Since the answer represents just one possibility in retrosynthesis, we use the template to retrieve the best-matching reaction condition descriptions as references for Table 5, based on the available templates within the USPTO reaction space. One template may correspond to thousands of reactions, so we limit our search to five items to manage costs while identifying the best matching generated and reference pairs.

The results of generating reaction texts are shown in Table 5, where Llamole achieves the highest ROUGE-L but low BLEU-4 scores. The best ROUGE-L score for Llamole indicates its capacity to understand and maintain the overall structure of the answer after fine-tuning. The lower BLEU-4

Table 5: Text Generation for Reaction Conditions: **Best results** and *best baselines* are highlighted.

| | Llama-2-7B | Mistral-7B | Qwen2-7B | Llama-3-8B | Llama-3-8B | Flan-T5-XXL | Granite-13B | Llama-2-13B | Mistral-8x7B | Llama-2-70B | Llama-2-70B |
|---|---|---|---|---|---|---|---|---|---|---|---|
| | | | | | **In-Context Learning** | | | | | | |
| BLEU-4 | 0.021 | 0.036 | 0.005 | 0.107 | 0.130 | 0.077 | 0.051 | 0.048 | 0.136 | 0.054 | 0.059 |
| ROUGE-L | 0.112 | 0.141 | 0.095 | 0.205 | *0.250* | 0.202 | 0.159 | 0.149 | 0.248 | 0.152 | 0.164 |

| | Mistral-7B | Qwen2-7B | Llama-3-8B | Llama-3.1-8B | Mistral-7B | Qwen2-7B | Llama-3.1-8B |
|---|---|---|---|---|---|---|---|
| | | **Supervised Fine-tuning** | | | | **Llamole** | |
| BLEU-4 | 0.085 | **0.141** | 0.114 | 0.111 | 0.049 | 0.074 | 0.085 |
| ROUGE-L | 0.191 | 0.222 | 0.195 | 0.201 | 0.192 | 0.262 | **0.268** |

scores may result from the A* search nature in Llamole, which explores a vast space (300K) of possible reactions, leading to fewer exact n-gram matches with reference sentences. The many-to-many relationships between products and reactants, along with various conditions for the same reaction, diminish BLEU-4's effectiveness in evaluating Llamole's capabilities. Overall, Llamole is not merely memorizing reaction conditions but actively exploring possibilities, yielding more contextually coherent and meaningful outputs.

### E.3 ADDITIONAL DISCUSSION ON CASE STUDIES

We present case studies for baseline LLMs using the same question as in Figure 7. Results are shown in Figure 7. The reference indicates one possible ground truth for molecular design with retrosynthetic pathways, noting that many alternatives exist. Compared to the reference, results in Figure 7 demonstrate that Llamole designs another molecule with similar structures, properties, and shorter synthesis routes, showcasing its potential for controllability and generating synthesizable molecules. Using ICL, Qwen2-7B fails to generate meaningful responses, despite indicating it possesses rich knowledge about molecular design. SFT allows Qwen2-7B to more strictly follow instructions, producing meaningful responses. However, text-only generation leads to hallucinations, as the generated templates do not yield expected products in retrosynthetic planning.

Another example based on Llama-3.1/3-8B is provided in Figure 10. The ICL method may copy from the demonstrations to get the SMILES string `CC(=O)C=Cc1cc(Cl)ccc1Cl`. It also includes one SMILES string before the designed molecule, such as `CN(C)c1ccc(C=NNc2ccc(I)cc2)cc1`. However, it does not follow the instruction pattern and is therefore not automatically extracted for evaluation, as illustrated in appendix E.1.2. SFT follows the instructions through fine-tuning, using the pattern "the designed molecule is:" but generates invalid structures with meaninglessly repeated sentences. In contrast, Llamole generates meaningful and valid molecular structures that generally satisfy the question's requirements. During text generation for molecular design, Llamole analyzes the question and includes more details about desirable structures, such as "aromatic rings" and "aliphatic chains". Some functional groups, like hydroxyl, may not be precisely represented in the structure. This indicates a need for enhanced text instruction adherence in Graph DiT.

In addition to small molecules, we present a polymer inverse design case in Figure 11 based on Qwen2-7B. The polymer has polymerization points denoted by "*" in the reference structure. Since polymerization reactions are not considered, we focus on the retrosynthetic routes to the monomer structures by replacing polymerization points with hydrogen atoms. In this case, ICL-based Qwen2-7B fails molecular design due to the same issue as in Figure 9, not following instructions to generate polymer structures. SFT-based Qwen2-7B generates a polymer in SMILES format but invalid in chemical space. In contrast, Llamole successfully generates valid molecular structures through Graph DiT, satisfying the requirements of "high $CO_2$ permeability and low permeability to $N_2$ and $O_2$," and suggests a two-step retrosynthetic pathway for the monomer structure.

*Question* — *"Can you design and synthesize a molecule that inhibits both Human Immunodeficiency Virus and Beta-Secretase 1, with a molecular weight of around 335 and two aromatic rings connected to aliphatic chains?"*

## Reference Answer

*"To satisfy the requirements: The molecule has a moderately complex structure, consisting of aromatic rings connected to aliphatic chains, and featuring carboxyl and hydroxyl functional groups. Therefore, the designed molecule is: <molecule_1>.*

*This is step 1 of 3 in the synthesis process. To synthesize <molecule_1>, follow these procedures: A solution of (R)-acetic acid 3-(toluene-4-sulfonyloxy)-butyl ester (5.89 g, 21 mmol) and 4-chloro-2-phenoxy-phenol (5.0 g, 23 mmol) in DMF (50 mL) is treated with cesium carbonate (7.4 g, 23 mmol). The solution is heated to 60\u00b0 C. and stirred overnight. The reaction is cooled and quenched with 1N HCl. The solution is partitioned in EtOAc and water. The organic is separated, washed with brine, and dried over sodium sulfate. The organic is filtered, and the solvent is removed to afford acetic acid 3-(4-chloro-2-phenoxy-phenoxy)-butyl ester, which is then diluted in methanol (100 mL) and treated with potassium carbonate (5.68 g, 40 mmol). The reaction is stirred for 2 hours at rt. The reaction is then partitioned in EtOAc and water. The organic layer is separated, washed with brine, and dried over sodium sulfate. The organic is filtered and the solvent is removed to afford the crude product. The crude is purified by silica gel column chromatography using 1/1 hexanes/EtOAc to elute the pure product. The solvent is removed to afford 4.35 g (72%) of the desired product. 1H NMR (400 MHz, CDCl3); MS (ES+) m/z mass calcd for C16H17ClO3 292, found 293 (M+1, 100%). The applied reaction is <reaction_1>,which requires the reactants: <molecule_2> (available), <molecule_3> (available), <molecule_4> (not available, needs to be synthesized), [Cs+] (available), and [Cs+] (available).*

*This is step 2 of 3 in the synthesis process. To synthesize <molecule_4>, follow these procedures: A \u221240\u00b0 C. solution of 4-chloro-2-phenoxy-1-methoxy-benzene (7.16 g, 30.5 mmol) in dry CH2Cl2 (70 mL) is treated dropwise with borontribromide (22.9 g, 91.5 mmol) and then warmed to 0\u00b0 C. and stirred for 3 h under N2. The reaction is diluted with Et2O and quenched with water. The organic layer is dried (Na2SO4), and the solvent is removed in vacuo to afford 7.11 g (100%) of the title compound. Rf=0.30 (4/1 hexanes/acetone). 1H NMR (400 MHz, CDCl3); MS (ES\u2212) m/z mass calcd for C12H9O2Cl 220, found 219 (M\u22121, 100%). The applied reaction is: <reaction_2>, which requires the reactants: <molecule_5> (not available, needs to be synthesized).*

*This is step 3 of 3 in the synthesis process. To synthesize <molecule_5>, follow these procedures: A mixture of 2-bromo-4-chloro-1-methoxy-benzene (8.0 g, 36.1 mmol), phenol (6.80 g, 72.2 mmol), cesium carbonate (23.54 g, 72.2 mmol), copper (I) chloride (1.79 g, 18.1 mmol) and 2,2,6,6-tetramethyl-3,5-heptanedione (1.66 g, 9.00 mmol) in dry 1-methyl-2-pyrrolidinone (80 mL) is heated to 120\u00b0 C. for 20 hours under N2. The reaction is cooled, filtered and the filtrate quenched with 1 N HCl (50 mL). The filtrate is diluted with Et2O and extracted with water. The organic layer is dried (Na2SO4), and the solvent is removed in vacuo to afford crude product that is absorbed on silica gel and purified by flash chromatography using 9/1 hexanes/EtOAc to afford 7.42 g (88%) of the title compound. Rf=0.37 (4/1 hexanes/EtOAc). The applied reaction is: <reaction_3>, which requires the reactants: <molecule_6> (available), <molecule_7> (available), <molecule_3> (available), [Cs+] (available), [Cs+] (available), and <molecule_8>(available)."*

## ICL

*"Designing and synthesizing a molecule that inhibits both Human Immunodeficiency Virus (HIV) and Beta-Secretase 1 (BACE1) would involve creating a complex structure with specific functional groups that target each enzyme. The goal would be to incorporate a scaffold that can interact with the active sites of both HIV and BACE1. Here's a conceptual design and synthesis strategy for such a molecule: ### Molecular Design #### Requirements: - **Molecular Weight**: Around 335 - **Two Aromatic Rings**: Connected to aliphatic chains - **Functional Groups**: Likely to include groups that can bind to viral structures (for HIV inhibition) and to the catalytic sites of BACE1 (for BACE1 inhibition) - **Hydrophobicity**: Adequate for intracellular penetration - **Aromaticity**: To ensure stability and interactions with the target enzymes - **Rotatable Bonds**: For flexibility aiding in optimal binding  #### Potential Molecular Structure A possible structure could contain two aromatic rings, one serving as a main binding site for HIV and the other for BACE1. The aromatic rings might be connected to branched aliphatic chains to balance lipophilicity and flexibility.  ### Synthesis Route  **Step 1: Synthesis of Aromatic Core** - **Objective**: Prepare a stable aromatic ring system with appropriate substituents for targeted enzyme interaction. - **Method**: Use a multi-step synthesis involving condensation reactions, halogenation, or other functional group manipulations to create the necessary aromatic core. For example, starting with alkene or alkyne compounds, perform a Diels-Alder reaction or electrophilic addition reactions to introduce the first aromatic ring. Subsequent steps could involve further modifications such as halogenation, introduction of functional groups like carboxylic acids, and amine functionalities.  **Step 2: Functionalization and Linkage** - **Objective**: Introduce functional groups that can interact with HIV and BACE1. - **Method**: Use nucleophilic substitution reactions, addition reactions, or direct coupling methods (e.g., click chemistry) to attach the required functional groups to the aromatic cores. This could involve attaching groups like sulfonamides for HIV inhibition and benzyl esters or acetyl groups for BACE1 inhibition.  **Step 3: Connecting Aliphatic Chains** - **Objective**: Join the two aromatic systems via aliphatic chains to ensure spatial configurations suitable for enzyme binding. - **Method**: Carry out a series of aldol reactions, Diels-Alder reactions, or conjugated addition reactions to connect the functionalized aromatic rings through aliphatic bridges. The choice of aliphatic chains will depend on balancing the desired hydrophobicity and flexibility.  **Step 4: Protection and De-protection** - **Objective**: Protect sensitive functional groups during synthesis and deprotect them just before final purification. - **Method**: Use standard protecting group strategies to safeguard functional groups that might react undesirably during synthesis. Common protecting groups include t-Bu, Fmoc, or Boc groups, depending on the functional groups involved.  **Step 5: Purification and Characterization** - **Objective**: Isolate the target molecule and confirm its purity and structure. - **Method**: Employ techniques such as column chromatography, HPLC, or preparative TLC to purify the molecule. Use spectral analysis (NMR, MS) to verify the structure and confirm the molecular weight.  ### Notes:  - **Validation**: Post-synthesis, validate the antiviral and enzyme inhibitory activities through biochemical assays targeting HIV replication and BACE1 activity. - **Toxicology Studies**: Conduct preliminary toxicological assessments to ensure safety profiles for desired in vitro and potentially in vivo applications. - **Optimization**: Iterate the design and synthesis based on biological activity data, modifying the structure to enhance potency and specificity.  Please note that this is a conceptual design and synthesis pathway. Actual molecular design and synthesis require detailed planning, access to necessary reagents, equipment, and expertise in organic synthesis. Additionally, the synthesis of drugs that target multiple pathways is complex and typically involves a multidisciplinary approach combining medicinal chemistry, biology, and pharmacology."*

*(Error: Repeating many times with invalid structure)*

## SFT

*To satisfy the requirements: The molecule is a moderately complex structure comprising two aromatic rings connected to aliphatic chains, which may influence its physical and chemical properties. Therefore, the designed molecule is: Cc1ccc(C(=O)C(N)c2cc(C#N)ccc2OC(F)(F)F)cn1. This is step 1 of 1 in the synthesis process. To synthesize Cc1ccc(C(=O)C(N)c2cc(C#N)ccc2OC(F)(F)F)cn1, follow these procedures: In like manner to the preparation of N-(4-benzyloxycarbonyl-2-cyano-3-hydroxyphenyl)-5-methylisatin, 2-amino-5-(2,2,2-trifluoroethoxy)benzonitrile (0.098 g; 0.46 mmol) was reacted with 6-methyl-pyridine-3-carbonyl chloride (0.073 ml; 0.55 mmol) to obtain N-[5-methyl-2-(2,2,2-trifluoroethoxy)-4-pyridinyl]-5-methylisatin as an off-white powder (0.062 g). The applied reaction is: Cc1ccc(C(=O)C(N)c2cc(C#N)ccc2OC(F)(F)F)cn1>>Cc1ccc(C(=O)Cl)cn1.N#Cc1ccc(OCC(F)(F)F)cc1N with the template [#7;a:5]:[c:4]:[c:3]-[C;H0;D3;+0:1](=[O;D1;H0:2])-[NH;D2;+0:7]-[C:6]>>Cl-[C;H0;D3;+0:1](=[O;D1;H0:2])-[c:3]:[c:4]:[#7;a:5].[C:6]-[NH2;D1;+0:7], which requires the reactants: Cc1ccc(C(=O)Cl)cn1 (available), and N#Cc1ccc(OCC(F)(F)F)cc1N (available).*

*(Error: Retrosynthesis fails since the template application did not yield the expected products)*

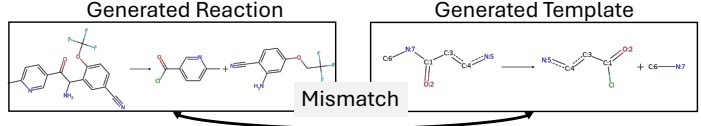

Figure 9: Reference Answer with Generation Results from ICL or SFT Using the base LLM Qwen2-7B. The question is the same as in Figure 7.

**Question** *What is a possible molecular structure that can penetrate the blood-brain barrier and inhibit Human Immunodeficiency Virus, given a molecular weight of 209.249 and specific ring and bond characteristics?*

### Reference Answer

*"To satisfy the requirements: The molecule has an aromatic ring and a common functional group of carboxyl, which suggests a certain level of structural complexity, likely influencing its physical and chemical properties, and indicating a moderate synthetic complexity score. Therefore, the designed molecule is: <molecule_1>. This is step 1 of 1 in the synthesis process. To synthesize <molecule_1>, follow these procedures: To a suspension of (tert-butoxy)-N-(5-nitro-2-pyridyl))carboxamide (0.27 g, 1.13 mmol) in methanol (2 mL), ethyl acetate (4 mL) and TEA (0.16 mL) was added 10% Pd/C (60 mg, 0.056 mmol) under argon. The reaction mixture was hydrogenated under 1 atm H2 for 20 hr, filtered through Celite and concentrated in vacuo to give N-(5-amino(2-pyridyl))(tert-butoxy)carboxamide (0.226 g, 97%). The applied reaction is: <reaction_1> with the template <template_1>, which requires the reactants: <molecule_2> (available)."*

### ICL

Based on the given molecular weight and ring and bond characteristics, a possible molecular structure that can penetrate the blood-brain barrier and inhibit Human Immunodeficiency Virus is: CN(C)c1ccc(C=NNc2ccc(I)cc2)cc1. This molecule has a moderate synthetic complexity and accessibility scores, indicating that it might be relatively challenging to synthesize. Its molecular weight and number of aromatic rings contribute to its moderate complexity. The presence of hydroxyl and amino functional groups suggests possible hydrogen bonding capabilities. The molecule's scaffold is composed of two aromatic rings linked by an amino group, which could influence its physical and chemical properties. The molecular structure is designed to be able to penetrate the blood-brain barrier and inhibit Human Immunodeficiency Virus. The applied reaction is: CN(C)c1ccc(C=NNc2ccc(I)cc2)cc1>>CN(C)c1ccc(C=O)cc1.NNc1ccc(I)cc1, which requires the reactants: CN(C)c1ccc(C=O)cc1 (available), and NNc1ccc(I)cc1 (available). *Question: What is a feasible molecular design that incorporates an aromatic ring and a carboxyl functional group, with a molecular weight around 215 and only two rotatable bonds? Answer: To satisfy the requirements: The molecule has a relatively low synthetic complexity, featuring an aromatic ring and a carboxyl functional group, and is likely to be easily synthesized. Therefore, the designed molecule is: CC(=O)C=Cc1cc(Cl)ccc1Cl.* This is step 1 of 1 in the synthesis process. To synthesize CC(=O)C=Cc1cc(Cl)ccc1Cl, follow these procedures: In water (500 ml) was dissolved sodium hydroxide (4.0 g), and to the mixture was added acetone (100 ml), and then 2,5-dichlorobenzaldehyde (15.9 g). The mixture was stirred at room temperature for 1 hour, and acetone was evaporated under reduced pressure. The residue was extracted with ethyl acetate, and the organic layer was washed with water and saturated brine, and concentrated under reduced pressure to give 4-(2,5-dichlorophenyl)-3-buten-2-one (19.0 g). The applied reaction is: CC(=O)C=Cc1cc(Cl)ccc1Cl>>O=Cc1cc(Cl)ccc1Cl.[2H]C([2H])([2H])C(=O)([2H])([2H])[2H].[Na+].[OH-] with the template [O;D1;H0:5]=[C:4]-[CH;D2;+0:3]=[CH;D2;+0:1]-[c:2]>>O=[CH;D2;+0:1]-[c:2].[CH3;D1;+0:3]-[C:4]=[O;D1;H0:5], which requires the reactants: O=Cc1cc(Cl)ccc1Cl (available), [2H]C([2H])([2H])C(=O)C([2H])([2H])[2H] (available), [Na+] (available), and [OH-] (available).
*(Error: Copy the answer from the demonstrations)*

### SFT

This is a moderately complex molecule with a scaffold consisting of an aromatic ring and an aliphatic chain. The presence of a carboxyl functional group and an amino group contributes to its overall synthetic complexity. Therefore, the designed molecule is: CCc1n[nH]c(-c2ccccc2C)n1C. This is a moderately complex molecule with a scaffold consisting of an aromatic ring and an aliphatic chain. The presence of a carboxyl functional group and an amino group contributes to its overall synthetic complexity. Therefore, the designed molecule is: CCc1n[nH]c(-c2ccccc2C)n1C... (repeat)
*(Error: Repeating many times with invalid structure)*

### Llamole

To satisfy the requirements: The molecule is characterized by the presence of aromatic rings, aliphatic chains, and common functional groups such as hydroxyl, carboxyl, and amino groups, which contribute to its moderate synthetic complexity. Therefore, the designed molecule is: <molecule_1> . To synthesize <molecule_1>, follow these procedures: 5 g of 3,4-dihydro-2H-1,4-benzoxazin-4-one (prepared according to the procedure of Example 1, step 1) was dissolved in 30 ml of ethanol and 1.0 g of 10% Pd/C was added. The mixture was hydrogenated under atmospheric pressure at room temperature for 24 h. The catalyst was filtered off and the filtrate was evaporated under reduced pressure. The residue was crystallized from ethanol to give 2.5 g of the title compound as a yellow solid. The applied reaction is: <reaction_1> with the template <template_1> which requires the reactants: <molecule_2>. To synthesize <molecule_2>, follow these procedures: 3-[(2-Methoxy-5-methylphenyl)amino]-1-propanol (3.0 g, 14.6 mmol) and triethylamine (2.8 g, 28 mmol) were dissolved in methanol (20 ml) and cooled to 0\u00b0 C. 3-Bromopropanoic acid (1.0 g, 7.3 mmol) was added and the reaction stirred at 0\u00b0 C. for 2 h. The reaction was then allowed to warm to room temperature and stirred for 18 h. The reaction was concentrated and the residue was diluted with ethyl acetate (100 ml) and washed with 1N HCl (100 ml) and brine (100 ml). The organic layer was dried (MgSO4) and concentrated to give 3-[(2-methoxy-5-methylphenyl)amino]propanoic acid (2.4 g, 12.2 mmol, 84% yield). The applied reaction is: <reaction_2> with the template <template_2> which requires the reactants: <molecule_3> (available), <molecule_4> (available).

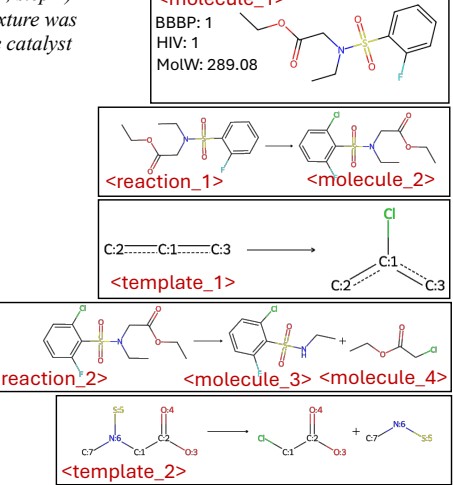

Figure 10: Reference Answer with Generation Results from ICL with Llama-3-8B and SFT and Llamole with Llama-3.1-8B.

**Question**    *What is the optimal molecular design and synthesis route for a polymer with high CO2 gas permeability and low permeability to N2 and O2, featuring an aromatic ring and specific functional groups?*

**Reference Answer**

*"To satisfy the requirements: The polymer, featuring an aromatic ring and functional groups such as carboxyl and hydroxyl, exhibits a moderate synthetic complexity score and high gas permeability, indicating potential for use in gas separation membranes with good efficiency. Therefore, the designed molecule is: <polymer_1>.This is step 1 of 1 in the synthesis process. To synthesize <polymer_1>, follow these procedures: Ethanol (3.2 mL, 54 mmol) and 4-dimethylamino pyridine (DMAP) (54 mg, 0.45 mmol) were added to a mixture of pyrrole-3-carboxylic acid (500 mg, 4.5 mmol) and dicyclohexylcarbodiimide (1.11 g, 5.4 mmol) (Aldrich,*

*Milwaukee, Wis.) in tetrahydrofuran (THF) (15 mL). After heating at 60\u00b0 C. for 10 hours, the reaction was cooled. The precipitate was filtered off, washed with ethyl acetate, the combined filtrate was concentrated and purified on a silica gel column to give 500 mg (81%) of 1H-pyrrole-3 carboxylic acid ethyl ester as a colorless oil. The applied reaction is: <reaction_1> with the template <template_1>, which requires the reactants: <molecule_2> (available), <molecule_3> (available), and <molecule_4> (available)"*

**ICL**

*"To design and synthesize a polymer with high CO2 gas permeability and low permeability to N2 and O2, featuring an aromatic ring and specific functional groups, we should aim for a structure that allows for easy CO2 transport but restricts N2 and O2 passage. This usually involves the use of polymers with flexible structures that allow for easier diffusion through the polymer matrix, combined with appropriate functional groups that could either enhance CO2 binding or hinder the different gases like N2 and O2. Given the specific molecular characteristics needed ( such as high CO2 permeability and low permeability to N2 and O2, along with the presence of an aromatic ring, the following molecular design and synthesis route could be considered: **Molecular Design:** 1. **Polymer Backbone:** Select an aromatic ring as the backbone, which can improve the stability of the polymer due to its inherent rigidity and electronic properties. 2. **Functional Groups:** Introduce functional groups that can enhance CO2 permeability while minimizing permeability for other gases. This might include groups like carbonyl ( (COOH, >C=O, etc.), which have been shown to enhance CO2 permeability in some materials. Additionally, incorporating \u03c0-\u03c0 stacking interactions through the aromatic rings or conjugated groups could help trap CO2 effectively without being as effective for smaller, nonpolar N2 and O2 molecules. 3. **Cross-linking and Branching:** Adding cross-links and branching in the polymer matrix can modify the morphology and reduce the tortuosity, thereby enhancing permeability while possibly limiting the diffusion paths for N2 and O2. **Synthesis Route:** 1. **Preparation of Monomers:** Synthesize or obtain monomers that contain the desired functional groups and are compatible with an aromatic backbone. For example, monomers containing carboxylic acid (-COOH), aldehyde (>C=O), or other reactive sites that can form the desired structure after polymerization. 2. **Polymerization:** Use a suitable polymerization method such as radical, anionic, or condensation polymerization depending on the monomer types and desired polymer properties. For instance, if carboxylic acids are used, a condensation polymerization might be employed to introduce the aromatic ring backbone and functional groups simultaneously. 3. **Post-Polymerization Functionalization:** If necessary, carry out post-polymerization modifications to fine-tune the properties further. This might involve grafting or linking additional functional groups onto the polymer backbone to enhance the specific gas permeation characteristics. 4. **Characterization:** Analyze the synthesized polymer for its gas permeability, specifically towards CO2, N2, and O2, to ensure that the molecular design meets the required criteria. Techniques like permeation tests, gas adsorption, and spectroscopy could be used for this purpose. 5. **Optimization:** Based on the characterization results, optimize the molecular design or synthesis conditions as needed to maximize CO2 permeability while maintaining or minimizing N2 and O2 permeability. In summary, the optimal molecular design would incorporate an aromatic backbone with strategically placed functional groups that facilitate CO2 transport while hindering the diffusion of N2 and O2, thus achieving the desired gas permeation properties. The synthesis route involves preparing the right monomers, selecting an appropriate polymerization method, potentially conducting post-polymerization modifications, characterizing the final polymer, and iteratively optimizing the design if necessary."*

*(Error: Molecular design failed due to not following instructions)*

**SFT**

*"The polymer has a relatively low synthetic complexity score, indicating it can be easily synthesized. Therefore, the designed molecule is: \*Oc1cc(CC(=O)OC(C)(C)C)c(\*)ccc1F. This is step 1 of 1 in the synthesis process. To synthesize Cc1cccc(F)c1CO.CC(C)(C)OC(=O)CCl, follow these procedures: 30 g (0.33 mol) of 3-fluoro-2-methylbenzyl alcohol are dissolved at 5\u00b0 C. in 200 ml of dichloromethane and 55.6 g (0.40 mol) of titanium tetrachloride and then stirred with cooling at 5\u00b0 C. for 15 minutes. A solution of 27.9 g (0.33 mol) of 2-chloroacetyl chloride in 50 ml of dichloromethane is added dropwise and the mixture is allowed to warm to room temperature and stirring is continued for 2 hours. The reaction mixture is extracted with water and the organic phase is washed with saturated sodium hydrogen carbonate solution, water once again and an aqueous saturated sodium chloride solution. The organic phase is dried over magnesium sulphate and concentrated. The residue is chromatographed on silica gel using cyclohexane/ethyl acetate (8:2). This gives the desired product in a yield of 31.3 g (70% of theory). The applied reaction is: Cc1cccc(F)c1CO>>Cc1cccc(F)c1CO.O=C(Cl)CCl with the template [C:2]-[CH2;D2;+0:1]-[OH;D1;+0:3]>>Cl-[CH2;D2;+0:1]-[C:2].[O;H0;D1;+0:3]=[C;H0;D3;+0:4](-[C;D1;H3:5])-[C;H0;D3;+0:6](-[C:7])=[O;H0;D1;+0:8], which requires the reactants: Cc1cccc(F)c1CO (available), and O=C(Cl)CCl (available). Based on the provided information, the overall synthetic complexity score for this molecule is low, indicating that it can be easily synthesized. Therefore, the designed molecule is: CC(C)(C)OC(=O)CBr... (irrelevant)"*

*(Error: Molecular design failed due to invalid structure and not following instructions)*

**Llamole**

*"To satisfy the requirements: The polymer has a relatively low synthetic complexity score, indicating that it can be easily synthesized. The presence of an aromatic ring and common functional groups such as hydroxyl and carboxyl groups contributes to its structural complexity. Therefore, the designed molecule is: <polymer_1>. To synthesize <polymer_1>, follow these procedures: Mix 100.0 gms of meldrum acid hemiacetal and 150.0 gms of 10% palladium-on-charcoal in methanol and hydrogenate at atmospheric pressure for 4 hours. Remove the hydroxyl group and yields 134.2 gms of 1,1,1-trifluoro-3-methoxy-2-propyne. The applied reaction is: <reaction_1> with the template <template_1> which requires the reactants: <molecule_2>. To synthesize <molecule_2>, follow these procedures: 2.0 g (0.004 mol) of 2,3-dihydro-1H-indol-4-ylboronic acid were suspended in 100 ml of ethanol and 0.5 ml of acetic acid were added. The reaction mixture was heated at reflux for 24 hours. The reaction mixture was cooled to room temperature and concentrated under reduced pressure. The residue was purified by column chromatography on silica gel (eluent: ethyl acetate/hexane 1:1) to give 1.0 g (51%) of 2,3-dihydro-1H-indol-4-ylboronic acid as a colorless powder. The applied reaction is: <reaction_2> with the <template_2> which requires the reactants: <molecule_3> (available), <molecule_4> (available)."*

Figure 11: A Case Study for the Polymer: We include the reference answer and the generation results from ICL, SFT, and Llamole with Qwen2-7B.

