# OpenReview forum: "Multimodal Large Language Models for Inverse Molecular Design with Retrosynthetic Planning"
_ICLR.cc/2025/Conference — ICLR 2025 Poster_

### Official Review · Reviewer_GFHU · 2024-11-02

**Soundness:** 4
**Presentation:** 4
**Contribution:** 3
**Rating:** 8
**Confidence:** 4

**Summary:**

Summary: In this paper, the authors address an important task of multi-modal molecule design using LLMs. In particular, they have integrated pre-trained Graph Models with pre-trained auto-regressive LLM and fine-tuned them using multi-modal data (Text+Graph) to enable interleaved generate Multi-modal output (text, molecules, and graph). Moreover, they have incorporated retrosynthetic planning during generation, where they integrate A* search to efficiently identify synthesis pathways for the designed molecule.

**Strengths:**

- The paper is very well written. The problem statement is well-motivated. The authors did a comprehensive review of prior work on LLMs for Molecule Synthesis.
- The authors proposed the first multi-modal LLM for molecular design. They proposed a novel idea to integrate Graph encoders with pre-trained LLMs and unified cross-entropy loss into the autoregressive modeling of LLMs.
- They curated a dataset and developed fine-tuning guidelines for benchmarking multimodal molecular design.
- Authors conduct extensive experiments with popular benchmark datasets and compare them with state-of-the-art models to show the effectiveness of their proposed model.
- Authors conduct ablation studies to show the utility of different components of their model.
- Source code is  provided

**Weaknesses:**

- Can the model handle any random format of the question, or is the format specified in Fig-2 predetermined?
- What is the minimum information required by LLMole to generate an appropriate molecular structure?
- To what extent does the information requested in the text match the generated material, and are there any qualitative results available for this?
- For property prediction for Drugs and Materials, Can the model handle unseen datasets for property prediction in drugs and materials? Could the authors provide some insights on this?

**Questions:**

Check the Weaknesses

---

> ### Author Response · Authors · 2024-11-19
>
> We sincerely appreciate the reviewer's thoughtful suggestions and questions. We have provided point-by-point answers to each weakness and question. We have also revised the main text and appendix to incorporate the reviewer's valuable feedback, with all changes clearly highlighted in blue for ease of reference. Should any concerns remain, we remain fully committed to addressing them promptly and thoroughly.
>
> ## W1: Random format of questions
>
> Thank you for the interesting question. As shown in Figure 2, Llamole is designed and fine-tuned to specialize in inverse molecular design tasks. It handles diverse question formats related to molecular design tasks (see Lines 1105-1113 for examples of question construction and case studies in Figures 7, 9, and 10).
>
> Llamole is based on 7B/8B LLMs, such as Llama-3.1 and Qwen-2, and is fine-tuned with LoRA which retains model quality for general tasks [1]. Larger base LLMs are generally expected to perform better with more varied question formats. For smaller LLMs, such as 7B models, this ability can still be enhanced by using a more diverse instruction dataset.
>
> ## W2: Minimum information required by llamole
>
> For inference, Llamole takes input questions specifying requirements for structures, properties, and synthesizability. No additional information, such as detailed structural or property instructions, is needed beyond what is provided in the question.
>
> For pretraining, Llamole requires a multimodal instruction tuning dataset. We curated around 128K instructions for efficient fine-tuning, which we demonstrate is sufficient for improved performance.
>
> "Appropriateness" can be defined from several aspects based on the question, such as whether the generated structure includes required functional groups (e.g., aromatic rings), satisfies desirable properties or is synthesizable through feasible pathways. These dimensions are evaluated in Tables 1 and 2, Figure 5, and the case studies in Figure 7.
>
> ## W3: Information match and results
>
> The structure similarity and drug/material property metrics highlight Llamole's advantage in generating molecules that meet the requirements specified in the question.
>
> Section 5.2.2 (Lines 456-462) and Appendix E.3 (Lines 1313-1332) discuss the qualitative results in Figures 7, 9, and 10, showing how the generated molecules match the specified properties and structural requirements. In summary, Figure 7 demonstrates that all key criteria specified in the question are satisfied.
>
> ## W4: Property prediction
>
> Llamole primarily focuses on inverse molecular design, not property prediction. However, property prediction is a promising future direction. In this case, we could train a text-based property predictor and adapt the multimodal autoregressive framework from Llamole by replacing the graph diffusion transformer with a pre-trained property prediction model. This would enable property prediction on unseen datasets.

---

> > ### Comment · Reviewer_GFHU · 2024-11-22
> > **Thank you for your Response.**
> >
> > Thank you for your thorough response. Your answers have addressed my concerns, and I’ve increased the rating score accordingly.

---

> > > ### Author Response · Authors · 2024-11-22
> > >
> > > We are glad to hear that your concerns have been addressed. Thank you once again for your thorough review and for raising the score!

---

### Official Review · Reviewer_iNVA · 2024-11-03

**Soundness:** 3
**Presentation:** 3
**Contribution:** 2
**Rating:** 6
**Confidence:** 2

**Summary:**

The paper introduces Llamole, a novel multimodal large language model (MLLM) designed for inverse molecular design that integrates graph and text generation capabilities for retrosynthetic planning. Llamole combines a base LLM with specialized graph modules (Graph Diffusion Transformer and Graph Neural Networks) to perform multi-conditional molecular generation and reaction inference within a unified framework. The model also incorporates A* search with LLM-based cost functions to enhance retrosynthetic planning efficiency. Extensive experiments demonstrate that Llamole outperforms other adapted LLMs on several benchmarks, establishing its effectiveness in controllable molecular design and retrosynthetic planning.

**Strengths:**

1. Novel Multimodal Integration: The integration of graph-based models with a base LLM to handle both text and graph data is innovative, addressing the limitations of LLMs in generating coherent molecular structures and enabling interleaved text and graph generation.

2. Efficient Retrosynthetic Planning: The use of A* search combined with LLM-derived cost functions significantly improves the efficiency of retrosynthetic planning, offering a practical solution for synthesizable molecular design.

3. Good Performance: Llamole achieves substantial improvements over baseline methods across multiple metrics, demonstrating its superiority in both molecular design and planning tasks.

**Weaknesses:**

1. Theoretical Justification: The paper lacks a strong theoretical explanation for why the Llamole approach works effectively for inverse molecular design. While the empirical results are compelling, a deeper theoretical discussion would enhance the understanding of its effectiveness.

2. Scalability Concerns: The computational cost and scalability of Llamole are not thoroughly discussed. The practicality of deploying such a large-scale multimodal model in real-world scenarios remains unclear, especially given the integration of multiple complex components.

3. Comparative Analysis Gaps: The paper does not compare Llamole with some recent multimodal approaches that also attempt to bridge the gap between graph and text modalities, which would provide a more comprehensive evaluation.

**Questions:**

See above

---

> ### Author Response · Authors · 2024-11-19
>
> We sincerely appreciate the reviewer's thoughtful suggestions and questions. We have provided point-by-point answers to each weakness and question. We have also revised the main text and appendix to incorporate the reviewer's valuable feedback, with all changes clearly highlighted in blue for ease of reference. Should any concerns remain, we remain fully committed to addressing them promptly and thoroughly.
>
> ## W1: Theoretical Justification
>
> Llamole builds on recent advances in generative models, including Transformers, (graph) diffusion models, and large language models (LLMs) [1, 2, 3, 4, 5, 6].
>
> Specifically, its theoretical foundation could draw from Transformer theory, such as in-context learning [1], LLMs' emergent abilities [2], parameter-efficient tuning [3], and diffusion models, including score-based models [4] and discrete diffusion models [5], which are well-suited for the discrete nature of molecular graphs [6]. We appreciate your recognition of the empirical performance. The theoretical understanding of LLMs, multimodal LLMs, and graph diffusion models is evolving; existing efforts provide a promising foundation for further theoretical exploration, which we believe is a valuable direction for future research.
>
> ## W2: Scalability concerns
>
> The models run efficiently on a single V100 for inference, completing molecular generation and retrosynthesis in 1 to 1.5 minutes. Specifically, the multi-conditional molecular generation takes about 40 seconds, including both text and molecular design. Users can control the maximum time for retrosynthetic planning, with a default value of 30 seconds.
>
> The primary cost comes from the large language models (7B to 8B parameters) used in this paper. Compared to LLM inference, we find that adding graph models (under 1B parameters) does not significantly increase the cost, either in terms of GPU or inference resources.
>
> ## W3: Comparative analysis gaps
>
> Thank you for your suggestion. We have added Appendix A to discuss more related work on multimodal language models and added Table 4 for further comparison. Below are the details:
>
> Existing multimodal approaches make valuable contributions to molecule-text tasks, such as molecular property prediction, captioning, and retrieval [7, 8, 9, 10, 11]. The task most similar to inverse molecular design is text-based molecular generation. In text-based molecular generation, multimodal language models [7, 10, 11] often use MolT5 [7] for molecular structure decoding. We evaluate all three variants of MolT5 (small, base, and large) for inverse molecular design, using a question as input to generate molecules. Results for drug design (averaged balanced accuracy across three tasks) are shown in the table below.
>
> | Model                    | MolT5-small (77M) | MolT5-base (250M) | MolT5-large (800M) | Best LLM | Llamole |
> |--------------------------|-------------------|-------------------|--------------------|----------|---------|
> | Averaged Accuracy                 | 0.150             | 0.232             | 0.264              | 0.502    | 0.662   |
>
> We find that the largest MolT5 still underperforms the best LLM (from ICL) for drug design. This may be due to the differences between text-based molecular generation (which takes descriptions of molecules as input) and inverse molecular design (which requires specific properties and synthesis path requirements with fewer details on the molecule itself).
>
> For material design, we find that MolT5 cannot generate valid polymer structures due to its lack of knowledge about polymerization points, typically represented by the asterisk symbol in SMILES strings. As a result, no valid MAE error is reported. Additionally, these models [7, 10, 11] have not addressed the retrosynthetic planning problem.
>
> ## Reference:
>
> [1] Understanding in-context learning in transformers and llms by learning to learn discrete functions. ICLR 2024.
>
> [2] Emergent abilities of large language models. TMLR. 2022.
>
> [3] Lora: Low-rank adaptation of large language models. ICLR 2022.
>
> [4] Score-Based Generative Modeling through Stochastic Differential Equations. ICLR 2021.
>
> [5] Structured Denoising Diffusion Models in Discrete State-Spaces. NeurIPS 2021.
>
> [6] Digress: Discrete denoising diffusion for graph generation. ICLR 2023.
>
> [7] Translation between Molecules and Natural Language. EMNLP 2022.
>
> [8] MolCA: Molecular Graph-Language Modeling with Cross-Modal Projector and Uni-Modal Adapter. EMNLP 2023.
>
> [9] GIMLET: A Unified Graph-Text Model for Instruction-Based Molecule Zero-Shot Learning. NeurIPS 2023.
>
> [10] GIT-Mol: A Multi-modal Large Language Model for Molecular Science with Graph, Image, and Text. Computers in biology and medicine. 2024.
>
> [11] MolFM: A Multimodal Molecular Foundation Model.

---

### Official Review · Reviewer_F4DQ · 2024-11-03

**Soundness:** 3
**Presentation:** 3
**Contribution:** 3
**Rating:** 6
**Confidence:** 5

**Summary:**

The paper introduces Llamole, a novel multimodal large language model (MLLM) designed for inverse molecular design. It is capable of autoregressive generation of interleaved text and graphs, which is a significant advancement in the field. Llamole integrates a base LLM with a Graph Diffusion Transformer and GNN model to enable molecular generation and molecular inference within textual contexts. Additionally, it incorporates an A* search algorithm with LLM-based cost functions for efficient retrosynthetic planning. The authors have further created benchmarking datasets, MolQA and MolPair, and conducted extensive experiments comparing Llamole against various state-of-the-art LLMs, demonstrating its superiority in controllable molecular design and retrosynthetic planning.

**Strengths:**

- The paper presents an autoregressive MLLM capable of understanding and generating interleaved text and graphs, which has the potential to be applied to a wide range of molecule-text downstream tasks.
- The authors have collected and processed new datasets, MolQA and MolPair, which allow for more complex molecule-text downstream evaluations. These datasets will be valuable resources for future research in the field if they can be made open-source.
- The experiments in controllable molecule generation are particularly compelling.
- The integration of the A* search algorithm with LLM-based cost functions is a practical and effective approach to retrosynthetic planning

**Weaknesses:**

- The framework of this work (Figure 3) is somewhat difficult to understand due to its tight layout and restricted color use. - Rearranging the layout and using a more varied color scheme could improve the clarity of the framework.
- The first claim about this paper's contribution is a bit vague. Although this model is indeed a novel MLLM (Multimodal LLM), there are already some existing MLLMs for molecular generation and understanding. The key innovation of this work lies in the autoregressive generation of interleaved text and graphs with GNN and Graph Diffusion Transformer. This should be emphasized in the introduction and preliminaries to distinguish it from existing work.
- As mentioned above, there are existing MLLM methods tailored for molecule-text related tasks, some of which can be scaled up to 7B or higher parameter levels. These models have made a great contribution to molecule discovery and should be included in the introduction (Figure 1), preliminaries, and experiment sections of this paper.
- This work conducts experiments majorly on the two new datasets proposed in this paper. While the comprehensiveness of the newly proposed datasets is commendable, additional comparisons on existing datasets such as MoleculeNet or USPTO retrosynthesis datasets would lead to better comparison with previous approaches.

**Questions:**

I've expressed all my concerns in the weaknesses session.

---

> ### Author Response · Authors · 2024-11-19
>
> We sincerely appreciate the reviewer's thoughtful suggestions and questions. We have provided point-by-point answers to each weakness and question. We have also revised the main text and appendix to incorporate the reviewer's valuable feedback, with all changes clearly highlighted in blue for ease of reference. Should any concerns remain, we remain fully committed to addressing them promptly and thoroughly.
>
> ## W1: Figure of the framework
>
> We have updated Figure 3 based on your feedback. Specifically, we added more text colors to emphasize different sections, highlighted the graph symbols in the text, and adjusted the layout to make it less compact, increasing the figure's width for improved clarity. If any concerns about the framework figure remain, we would be happy to address them and continue refining it.
>
> ## W2: Claim about contribution
>
> Thank you for your comment. We have updated the claim in the last paragraph of the introduction to: "Llamole is the first MLLM capable of inverse molecular design with the interleaved generation of text and graphs" for better clarity.
>
> We acknowledge the valuable contributions of multimodal language models and have provided further details in the related work section (Lines 516-518) and Appendix A (Lines 866-892). In contrast to these related work, this work focuses on inverse molecular design, emphasizing (1) multi-conditional and (2) synthesizable molecular generation, areas that have been underexplored in prior work.
>
> ## W3: Related work
>
> We appreciate your suggestions and have clarified the relationship to existing multimodal language models in the related work section (Lines 516-518) and Appendix A (Lines 866-892). We also provide a new comparison in Table 4. Below are the details:
>
> Existing multimodal language models make valuable contributions to molecule-text tasks, such as molecular property prediction, captioning, and retrieval [1, 2, 3, 4, 5]. The task most similar to ours (inverse molecular design) is text-based molecular generation. These multimodal language models [1, 4, 5] are often based on MolT5 [1] for molecular structure decoding. Thus, we evaluate them in new experiments. We test all three variants of MolT5 (base, small, and large) for text-based molecular generation, using a question as input and asking the models to generate molecules. Results for drug design (averaged balanced accuracy across three tasks) are provided in the table below.
>
> | Model                    | MolT5-small (77M) | MolT5-base (250M) | MolT5-large (800M) | Best LLM | Llamole |
> |--------------------------|-------------------|-------------------|--------------------|----------|---------|
> | Averaged Accuracy                 | 0.150             | 0.232             | 0.264              | 0.502    | 0.662   |
>
> We find that the largest MolT5 still underperforms the best LLM (from ICL) for drug design. This may be due to the differences between text-based molecular generation (which takes descriptions of molecules as input) and inverse molecular design (which requires specific properties and synthesis path requirements with fewer details on the molecule itself).
>
> For material design, we find that MolT5 cannot generate valid polymer structures due to its lack of knowledge about polymerization points, typically represented by the asterisk symbol in SMILES strings. As a result, no valid MAE error is reported. Additionally, these models [1, 4, 5] have not addressed the retrosynthetic planning problem.
>
> ## W4: Existing datasets
>
> Thank you for your suggestions. The mentioned datasets already contribute to the existing comparisons. Relevant details can be found in Lines 291-293, Figure 4 of Section 4, with further information in Appendix C. Below is a summary.
>
> We have constructed new datasets for the comprehensive evaluation of question answering in inverse molecular design. Key data sources include MoleculeNet and USPTO, as shown in Figure 4. MoleculeNet properties, such as HIV virus replication inhibition (HIV), β-secretase 1 inhibition (BACE), and blood-brain barrier permeability (BBBP), are used for evaluating multi-conditional molecular generation. We also incorporate small molecules from ChEMBL, PubChem, and ZINC, as well as polymers from various sources. The 3.8 million records from USPTO are used for GNN pre-training, Llamole fine-tuning, and evaluation.
>
> Thank you again for your valuable feedback. We are happy to address any further concerns and welcome additional discussion.
>
> ## Reference
> [1] Translation between Molecules and Natural Language. EMNLP 2022.
>
> [2] MolCA: Molecular Graph-Language Modeling with Cross-Modal Projector and Uni-Modal Adapter. EMNLP 2023.
>
> [3] GIMLET: A Unified Graph-Text Model for Instruction-Based Molecule Zero-Shot Learning. NeurIPS 2023.
>
> [4] GIT-Mol: A Multi-modal Large Language Model for Molecular Science with Graph, Image, and Text. Computers in biology and medicine. 2024.
>
> [5] MolFM: A Multimodal Molecular Foundation Model.

---

### Official Review · Reviewer_DEua · 2024-11-09

**Soundness:** 3
**Presentation:** 2
**Contribution:** 3
**Rating:** 8
**Confidence:** 2

**Summary:**

This paper propose a multimodal large language model framework to jointly model molecules and texts for drug discovery and retrosynthesis planning tasks, which is motivated by the fact that the joint modeling of texts and molecules remain challenging.  To address this, the framework combines an existing LLM with two pre-trained graph encoders: one for conditional molecule generation and another for predicting reaction templates for retrosynthesis planning. The paper further proposes to apply A* search to ensure the target product could be synthesized from reactants that are purchasabe on the market. To summarize, the goal of this paper is to propose a MLLM-based framework that is able to build a pipeline from drug design to retrosynthesis planning for the discovered molecules. To evaluate the proposed framework, the paper proposes a dataset to benchmark in such task. From the empirical experiments, the paper argues that the proposed framw3ork Llamole outperforms the baselines of ICL or SFT with only LLMs, as well as GraphGA. The paper also justifies their deisgn of the framework based on ablation studies.

**Strengths:**

1. The idea proposed in the paper on jointly model texts and molecules with MLLM for inverse molecular deisgn and retrosynthesis planning pipeline is novel. The created dataset could also contribute to the area;
2. From the experiments the proposed framework shows strong performance.
3. The paper is clearly written for readers to understand the proposed method.

**Weaknesses:**

The baselines selected in the experiments are mainly LLMs with ICL or SFT. The only baseline with similar purpose is GraphGA. It would be interested to see the comaprisons of more baselines of the combination of existing inverse molecular design [1] and retrosynthesis planning [2,3] methods.

1. Weiss, Tomer, et al. "Guided diffusion for inverse molecular design." Nature Computational Science 3.10 (2023): 873-882.
2. Zeng, Tao, et al. "Developing BioNavi for Hybrid Retrosynthesis Planning." JACS Au 4.7 (2024): 2492-2502.
3. Han, Peng, et al. "Gnn-retro: Retrosynthetic planning with graph neural networks." Proceedings of the AAAI conference on artificial intelligence. Vol. 36. No. 4. 2022

**Questions:**

What is the computational cost and inference time of the proposed methods, compared to GraphGA?

---

> ### Author Response · Authors · 2024-11-19
>
> We sincerely appreciate the reviewer's thoughtful suggestions and questions. We have provided point-by-point answers to each weakness and question. We have also revised the main text and appendix to incorporate the reviewer's valuable feedback, with all changes clearly highlighted in blue for ease of reference. Should any concerns remain, we remain fully committed to addressing them promptly and thoroughly.
>
> ## W1 Baseline comparisons
>
> Thank you for your suggestions. We have updated Section 6 (Related Work) to discuss these methods. We have updated Tables 1 and 2 in the main text to include the new results. We found these baselines interesting, but challenging to reproduce and apply to the tasks in this work.
>
> First, GaUDI [1] is a diffusion model that uses additional predictors to guide molecular generation. However, both the diffusion model and predictor require accurate 3D structures for training. Following the instructions in their Method section [1], obtaining these structures involves several extensive steps, including initial estimation (RDKit), pre-optimization (Riniker [4] + UFF [5]), optimization (GFN2-xTB), filtering, and post-processing. These steps can take weeks or even months. We believe that developing and evaluating multimodal large language models for 3D inverse molecular design is a promising direction, but substantial work remains, which we may explore in future studies.
>
> Second, we were unable to find the code for [3], which made it difficult to reproduce and apply their method to the retrosynthesis task in this paper within a short time frame.
>
> As an alternative, we used the recently proposed DiGress with additional predictor guidance [6], combined with BioNavi [2] (with a maximum planning time of 30s, same as Llamole), to integrate inverse molecular design and retrosynthesis planning. We trained the model using molecules and properties from the MolQA training set and tested the generation tasks on the test set. Comparison results are shown in the table below.
>
> | Model                       | Validity | Similarity | HIV BA   | BBBP BA  | BACE BA  | Overall BA | Drug Synthesis Success Rate (%) | CO2Perm MAE | N2Perm MAE | O2Perm MAE | FFV MAE   | TC MAE    | Overall MAE | Material Synthesis Success Rate (%) |
> |-----------------------------|----------|------------|-------|-------|--------|------------|----------------------------|---------|--------|--------|-------|-------|-------------|----------------------------|
> | DiGress+BioNavi             | 0.375    | 0.046      | 0.515 | 0.522 | 0.580  | 0.539      | 18                         | 0.655   | 1.884  | 0.680  | 0.020 | 0.049 | 0.657       | 15.4                       |
> | GraphGA                     | 0.885    | 0.112      | 0.536 | 0.515 | 0.560  | 0.537      | NA                         | 0.847   | 1.556  | 0.747  | 0.020 | 0.042 | 0.642       | NA                         |
> | Llamole (Llama3.1)          | 0.913    | 0.142      | 0.623 | 0.629 | 0.713  | 0.655      | 35.1                       | 0.653   | 1.344  | 0.549  | 0.021 | 0.030 | 0.519       | 17.6                       |
>
> Neither DiGress+BioNavi nor GraphGA can generate text to describe the molecular design process. DiGress+BioNavi shows lower validity but performs comparably to GraphGA in generating molecules that satisfy property conditions. Llamole outperforms DiGress+BioNavi in designing synthesizable molecules, with a higher success rate in finding synthesis paths.
>
> ## Q1: Computational cost
>
> The models run efficiently on a single V100 for inference, completing molecular generation and retrosynthesis in 1 to 1.5 minutes. Specifically, the multi-conditional molecular generation stage takes about 40 seconds, including both text and molecular design. Users can control the maximum retrosynthesis time, with a default of 30 seconds, ensuring the model returns a result within that time, regardless of success or failure. Users can also adjust the timing for extended planning if needed. In comparison, GraphGA completes molecular generation in around 15 seconds but lacks text generation and retrosynthetic planning capabilities.
>
> ## Reference
>
> [1] Guided diffusion for inverse molecular design. Nature Computational Science. 2023
>
> [2] Developing BioNavi for Hybrid Retrosynthesis Planning. JACS 2024.
>
> [3] Gnn-retro: Retrosynthetic planning with graph neural networks. AAAI 2022.
>
> [4] Better Informed Distance Geometry: Using What We Know To Improve Conformation Generation. JCIM 2015.
>
> [5] UFF, a full periodic table force field for molecular mechanics and molecular dynamics simulations. JACS 1992.
>
> [6] DiGress: Discrete Denoising diffusion for graph generation. ICLR 2023.

---

> > ### Comment · Reviewer_DEua · 2024-11-19
> > **Thank you for your response.**
> >
> > Thank you for your responses. These responses resolve my concerns.

---

> > > ### Author Response · Authors · 2024-11-19
> > > **Thank you for your prompt response**
> > >
> > > We are delighted to hear that your concerns have been properly addressed and that the scores have been raised. Thank you once again for your valuable and helpful feedback. We warmly welcome any new discussions.

---

### Meta-Review · Area_Chair_nJLE · 2024-12-21

**Metareview:**

This paper proposes a multimodal large language model for inverse molecule design and retrosynthetic analysis.

The method is well-designed and the experiments are well-executed. The paper also proposes new datasets and benchmarks which are likely to be very useful for future research in this area.

Reviewers pointed out some presentation issues and missing baselines, but the concerns were not critical.

Overall, this is a very solid work. I recommend acceptance.

**Additional Comments On Reviewer Discussion:**

The reviewers pointed out some concerns on the presentation and the experiments (lack of baselines, comparisons in existing setups), but they are not critical enough to harm the main claims and contribution of the paper.

---

### Decision · Program_Chairs · 2025-01-22

Accept (Poster)